# Huge Ensembles Part I: Design of Ensemble Weather Forecasts using Spherical Fourier Neural Operators

Ankur Mahesh<sup>1,2,\*</sup>, William D. Collins<sup>1,2,\*</sup>, Boris Bonev<sup>3</sup>, Noah Brenowitz<sup>3</sup>, Yair Cohen<sup>3</sup>, Joshua Elms<sup>4</sup>, Peter Harrington<sup>5</sup>, Karthik Kashinath<sup>3</sup>, Thorsten Kurth<sup>3</sup>, Joshua North<sup>1</sup>, Travis O'Brien<sup>4</sup>, Michael Pritchard<sup>3,6</sup>, David Pruitt<sup>3</sup>, Mark Risser<sup>1</sup>, Shashank Subramanian<sup>5</sup>, and Jared Willard<sup>5</sup>

Correspondence: Ankur Mahesh (ankur.mahesh@berkeley.edu)

#### Abstract.

Simulating low-likelihood high-impact extreme weather events in a warming world is a significant and challenging task for current ensemble forecasting systems. While these systems presently use up to 100 members, larger ensembles could enrich the sampling of internal variability. They may capture the long tails associated with climate hazards better than traditional ensemble sizes. Due to computational constraints, it is infeasible to generate huge ensembles (comprised of 1,000-10,000 members) with traditional, physics-based numerical models. In this two-part paper, we replace traditional numerical simulations with machine learning (ML) to generate hindcasts of huge ensembles. In Part I, we construct an ensemble weather forecasting system based on Spherical Fourier Neural Operators (SFNO), and we discuss important design decisions for constructing such an ensemble. The ensemble represents model uncertainty through perturbed-parameter techniques, and it represents initial condition uncertainty through bred vectors, which sample the fastest growing modes of the forecast. Using the European Centre for Medium-Range Weather Forecasts Integrated Forecasting System (IFS) as a baseline, we develop an evaluation pipeline composed of mean, spectral, and extreme diagnostics. Using With large-scale, distributed SFNOs with 1.1 billion learned parameters, we achieve calibrated probabilistic forecasts. As the trajectories of the individual members diverge, the ML ensemble mean spectra degrade with lead time, consistent with physical expectations. However, the individual ensemble members' spectra stay constant with lead time. Therefore, these members simulate realistic weather states during the rollout, and the ML ensemble thus passes a crucial spectral test in the literature. The IFS and ML ensembles have similar Extreme Forecast Indices, and we show that the ML extreme weather forecasts are reliable and discriminating. hese diagnosticsy These diagnostics ensure that the ensemble can reliably simulate the time evolution of the atmosphere, including low likelihood high-impact extremes. In Part II, we generate a huge ensemble initialized each day in summer 2023, and we characterize the statistics simulations of extremes.

<sup>&</sup>lt;sup>1</sup>Earth and Environmental Sciences Area, Lawrence Berkeley National Laboratory (LBNL), Berkeley, California, USA

<sup>&</sup>lt;sup>2</sup>Department of Earth and Planetary Science, University of California, Berkeley, USA

<sup>&</sup>lt;sup>3</sup>NVIDIA Corporation, Santa Clara, California, USA

<sup>&</sup>lt;sup>4</sup>Department of Earth and Atmospheric Sciences, Indiana University, Bloomington, Indiana, USA

<sup>&</sup>lt;sup>5</sup>National Energy Research Scientific Computing Center (NERSC), LBNL, Berkeley, California, USA

<sup>&</sup>lt;sup>6</sup>Department of Earth System Science, University of California, Irvine, USA \*These authors contributed equally to this work.

#### 1 Introduction

Recent low-likelihood, high-impact events (LLHIs) have raised important and unanswered questions about the drivers of these events and their relationship to anthropogenic climate change. For example, Hurricane Harvey in 2017 and the Summer 2021 heatwave in the Pacific Northwest (PNW) are two high-impact events with no modern analog. Several threads motivate research on LLHIs. First, the IPCC states that "the future occurrence of LLHI events linked to climate extremes is generally associated with *low confidence*" (Seneviratne et al., 2021, pp. 1536). Second, the occurrence of recent LLHIs, such as as-the Summer 2021 PNW heatwave, reveals that the abilityour abilities to characterize, let alone anticipate, such events is are currently incomplete (Bercos-HickeyBercosHickey et al., 2022; Zhang et al., 2024; Liu et al., 2024).

LLHIs challenge the standard climate models that might be used to answer such questions. Computational costs make it infeasible to run the large ensembles of simulations that are necessary to make inferences about the statistics of extremely rare weather events. The climate modeling community has successfully constructed large ensembles of up to  $O(10^2)$  members, such as the Community Earth System Model 2 Large Ensemble (CESM2-LE). To examine the rarest of LLHIs, a larger sample size is necessary. For instance, McKinnon and Simpson (2022) note, "for very large events (e.g., exceeding  $4.5\sigma$  at a weather station), only a small minority of CESM2-LE analogs in skewness/kurtosis space produce similarly extreme events."

These challenges motivate the application of entirely new methodological approaches, such as those based on machine learning (ML). For the first time, it is now possible to generate massive ensembles using ML with orders-of-magnitude less computational cost than traditional numerical simulations (Pathak et al., 2022). Recent work has demonstrated the potential of deterministic ML-based weather forecasting, which has comparable or superior root-mean squared error (RMSE) to the Integrated Forecasting System (IFS) at 0.25 degree horizontal resolution (Bi et al., 2023; Lam et al., 2023; Willard et al., 2024; ECMWF). Olivetti and Messori (2024) show that these deterministic data-driven models also offer promising forecast skill on extremes, and Pasche et al. (2024) validate them on case studies of extreme events. As our ML architecture, we use Spherical Fourier Neural Operators (SFNO) (Bonev et al., 2023). SFNO has been proven to be efficient and powerful in modeling a wide range of chaotic dynamical systems, including turbulent flows and atmospheric dynamics, while remaining numerically stable over long autoregressive rollouts. Given these promising deterministic results, we use ML to create ensemble forecasts, which provide probabilistic weather predictions. A high-level design decision is whether to create the ensemble after training the ML model or during the training itself. We use the former approach: we train ML models to minimize the deterministic mean squared error (MSE) at each time step. After training, we create a calibrated ensemble by representing initial condition and model uncertainty. Conversely, NeuralGCM (Kochkov et al., 2023), FuXi-ENS (Zhong et al., 2024), SEEDS (Li et al., 2024), and GenCast (Price et al., 2023) employ probabilistic training objectives instead of deterministic RMSE.

In this two-part paper, we present a first-of-its-kind huge ensemble of weather extremes using an ML-based emulator of global numerical reanalyses. In Part I, we introduce the ML architecture and the ensemble design (Section 2). In Table 1, we list the major design decisions of the ensemble, and we include pointers to the relevant sections in the paper for understanding the decision-making criteria. We benchmark the ML performance against an operational weather forecast, the

European Center for Medium-range Weather Forecast's (ECMWF) Integrated Forecasting System ensemble (IFS ENS). We assess whether our ML ensemble is fit for purpose using a suite of diagnostics that assess the overall probabilistic performance of the ensemble.

Figure 1. Overview of ensemble architecture. The ensemble is constructed using two methods: initial condition perturbations and model perturbations. The initial condition perturbations are generated using bred vectors, to sample the fastest growing errors in the initial condition. Model perturbations consist of twenty-nine instances of the SFNO model trained independently from scratch. Bred vectors are generated separately for each SFNO checkpoint. Each bred vector creates two initial condition perturbations: one with the bred vector added to the initial condition, and one with the bred vector subtracted from the initial condition. For the small ensemble, we use N = 1 bred vectors per checkpoint. For the huge ensemble in Part II, there are N = 128 bred vectors per checkpoint.

ML ensemble is fit for purpose using a suite of diagnostics that assess the overall probabilistic performance of the ensemble and its spectra. Because of our interest in LLHIs, we also present an extremes diagnostics pipeline that specifically assesses MLensemble extreme weather forecasts. In Part II<sub>7</sub> (Mahesh et al., 2024), we analyze a huge ensemble hindcast with hindcasts of 7,424 ensemble members.

#### 2 Designing ensembles with SFNO

We adopt the SFNO training scheme presented in Bonev et al. (2023). SFNO is trained on the European Center for Mediumrange Weather Forecasts Reanalysis v5 (ERA5) (Hersbach et al., 2020) at the dataset's 0.25-degree horizontal resolution. The weights of SFNO are optimized to minimize the latitude-weighted deterministic MSE loss function. Each model is trained for When calculating the loss, each variable is weighted by pressure and by the time tendency of the variable in the training dataset, similar to methods presented in Lam et al. (2023) and Watt-Meyer et al. (2023).

#### 70 epochs

To create ensemble weather forecasts with SFNO, we mirror methods used in numerical weather prediction. Figure 1 provides an overview of our ensemble generation method. For weather forecasts, two major sources of uncertainty are initial condition uncertainty and model uncertainty. Initial condition uncertainty stems from the inaccuracies in observing the current meteorological state, while model uncertainty arises from the incompletely known and imperfect numerical representations of physics that govern the atmosphere's time evolution. To represent initial condition uncertainty, we use bred vectors, a method formerly used by the Global Ensemble Forecast System (GEFS) (Toth and Kalnay, 1993, 1997). Bred vectors are designed to sample the fastest growing directions of the ensemble error patterns. By creating rapidly diverging ensemble trajectories, bred vectors are designed to create an ensemble that fully represents the probability of future weather states. Existing work has shown that simple Gaussian perturbations do not yield a sufficiently dispersive ensembleML ensembles (Scher and Messori, 2021; Bülte et al., 2024): the ensemble spread from these perturbations is too small. Bred

vectors <u>solvehelp address</u> this problem by creating a more dispersive ensemble which better reflects the full distribution of possible future states of the atmosphere. They <u>agnostically</u> amplify the fastest growing modes in <u>athe</u> model's intrinsic dynamics. While bred vectors have been used to create ensemble forecasts from traditional dynamical models, assessing how ML models respond to <u>such</u> perturbations is an important research frontier.

To represent model uncertainty, we train multiple SFNO models from scratch. We refer to each trained SFNO instance as a "checkpoint." At the start of training, each checkpoint is initialized with different random weights. During training, SFNO iteratively updates its weights to minimize a loss function: in this case, the loss function is the mean-squared error between the model predictions and the ERA5 training data. During each epoch of training, SFNO iterates through the entire training dataset and updates its weights to minimize the loss. We train SFNO for 70 total epochs. By the end of training, the models converge to a different local optimum of learned weights. The resulting ensemble of the different trained SFNO checkpoints represents the uncertainty in the SFNO model weights itself. Each resulting checkpoint represents an equivalently plausible set of weights that can model the time evolution of the atmosphere from an initial state. With multiple checkpoints, we create an ensemble with a spread of forecasts, yet each ensemble member has the same skill. Weyn et al. (2021) use multiple checkpoints to create an ensemble of forecasting models for medium-range and subseasonal prediction. They reduce computational costs by saving multiple model checkpoints from each training run and training the last few epochs independently for each model. This approach requires several additional design decisions: how should the learning rate for the optimization during these last retrained epochs be adjusted? How many extra epochs should each checkpoint train for? At what point during training should the checkpoints diverge? To minimize the ensemble's dependence on these hyperparameters, we opt to retrain each checkpoint completely from scratch.

We create an ensemble called SFNO-BVMC: Spherical Fourier Neural Operators with Bred Vectors and Multiple Checkpoints. In Table 1, we present a list of hyperparameters and their associated criteria that we use to guide our choice of ensemble design. We use a train-validation-test set paradigm. SFNO is trained on the years 1979-2016. We use the year 2018 as a validation year, on which we tune multiple aspects of the ensemble, such as the amplitude of the bred vectors and the number of SFNO checkpoints. Because these ensemble parameters are tuned using the year 2018, we cannot use 2018 for unbiased evaluation of the final ensemble. For our overall diagnostics, the year 2020 is used as an out-of-sample, held-out test set This year is used in the test set on the WeatherBench 2 (Rasp et al., 2024) platform, allowing for simplified comparison of SFNO-BVMC with other ML-based ensemble weather forecasting systems. To evaluate the skill for extreme weather, we use boreal summer 2023 (June, July, August) because it iswas the hottest summer in recorded history at the time (Esper et al., 2024). In Part II, we present a deep dive on a huge ensemble of forecasts from this

Table 1. Ensemble Design Decisions. A list of ensemble design decisions used to create the ML ensemble. The pointer to the section in the paper includes a more in-depth explanation of each decision and the criteria for making the choice.

| Name         | Value                                     | Paper Section       |
|--------------|-------------------------------------------|---------------------|
| Architecture | Spherical Fourier Neural Operators v0.1.0 | Part I, Section 2.1 |

| Training Dataset                                                     | 1979-2015                                                                                                                        | Part I, Section 2   |
|----------------------------------------------------------------------|----------------------------------------------------------------------------------------------------------------------------------|---------------------|
| Validation Dataset                                                   | 2018                                                                                                                             | Part I, Section 2   |
| Test Dataset                                                         | 2020                                                                                                                             | Part I, Section 2   |
| Forecast Time Step                                                   | 6 hours                                                                                                                          | Part I, Section 2   |
| Horizontal Resolution                                                | 0.25 degrees                                                                                                                     | Part I, Section 2   |
| Embedding Dimension                                                  | 620                                                                                                                              | Part I, Section 2.1 |
| Scale Factor                                                         | 2                                                                                                                                | Part I, Section 2.1 |
| Autoregressive Fine-tuning                                           | None                                                                                                                             | Part I, Section 3.2 |
| Training Time                                                        | 16 hours on 256 A100 GPUs per checkpoint                                                                                         | Part I, Section 2.1 |
| Inference Time                                                       | 1 second per 6 hour timestep on 1 NVIDIA A100 GPU                                                                                | Part I, Section 2.1 |
| Variable Set                                                         | 73 channels from Bonev et al. (2023) and 2m dewpoint temperature.  The pressure variables are represented on 13 pressure levels. | Part I, Section 2.1 |
|                                                                      | The pressure variables are represented on 15 pressure levels.                                                                    |                     |
| Bred Vector Amplitude                                                | 0.35 * SFNO Deterministic RMSE at 48 hours                                                                                       | Part I, Section 2.3 |
| Centered Bred Vectors                                                | Each bred vector is added to and subtracted from the initial condition                                                           | Part I, Section 2.3 |
| Hemispheric Rescaling for Bred<br>Vectors                            | Perturbations are rescaled separately polewards of 20 degrees. A linear interpolation is used for rescaling in the tropics.      | Part I, Section 2.3 |
| Initial Noise for Bred Vectors                                       | Adding spherical noise (correlated on 500 km length scales) to z500                                                              | Part I, Section 2.3 |
| Number of Model Checkpoints                                          | 29                                                                                                                               | Part I, Section 2.2 |
| Number of Perturbations per Model<br>Checkpoint (Benchmark Ensemble) | 2, with 1 centered (1 bred vector perturbation that is added to and subtracted from the                                          | Part I, Section 3   |
|                                                                      | initial condition)                                                                                                               |                     |
| Number of perturbations per Model<br>Checkpoint (Huge Ensemble)      | 256 <del>, with (128 centered bred vector perturbations, each added to and subtracted from the initial condition)</del>          | Part II             |
| Total size of huge ensemble                                          | 7424 members                                                                                                                     | Part II             |
| Lead Time to Analyze Extreme Statistics                              | <del>3 - 5</del> 2 days, 4 days, 10 days                                                                                         | Part I, Section 3.3 |
| Derived Variables in Huge Ensemble                                   | Integrated Vapor Transport, 10m wind speed, heat index                                                                           | Part II             |

dive on a huge ensemble of forecasts from this time period. No SFNO training or ensemble design decisions were made using the year 2020 or 2023. This setup with different training, validation, and test sets is crucial to avoid data leakage.

# 2.1 Selecting an emulator

SFNO is an ML architecture built on neural operators (Li et al., 2020), which are designed to learn mappings between function spaces. They can be used for different discretizations and grids, and they have broad applicability to various partial differential equation (PDE) problems. SFNO is a special instance of a Neural Operator, which uses the Spherical Harmonic Transform to represent operators acting on functions defined on the sphere. The spherical formulation leads to a strong inductive bias, respecting underlying symmetries the geometry and symmetry of the problems phere. This reduces error buildup and leads to stableduring autoregressive rollouts, making the methods ideally suited for PDE problems on the sphere. We use the open-source version of SFNO v0.1.0 released in the modulus-makani Python repository (Bonev et al., 2024).

We provide a brief overview of the SFNO architecture; for a more detailed explanation, refer to Bonev et al. (2023). The SFNO architecture consists of three main components: the encoder, the SFNO blocks, and the decoder.

- 1. Encoder: The encoder employs multi-layer perceptrons (MLPs) at each grid cell to map the input fields into a higherdimensional latent space. MLPs are fully connected neural networks that apply nonlinear transformations to their inputs.
- 2. SFNO Blocks: The processor incorporates 8 SFNO blocks, operating in the latent space, each performing two main operations:
  - (a) A spherical convolution with a learned filter encoded in the spectral domain. The signal is transformed into the spectral domain and back via a spherical harmonic transform (SHT) and its inverse. In the spectral domain, the convolution operation becomes a pointwise multiplication.
  - (b) An MLP applies nonlinear transformations to the latent features.

The output of each SFNO block serves as the input to the subsequent block. The first block downsamples the input resolution by a specified "scale factor," while the last block upsamples back to the original resolution.

3. Decoder: The decoder maps the latent space back into physical space using MLPs.

SFNO encodes an operator that maps functions defined on the sphere to other functions on the sphere. This learned map is parameterized by the weights of the MLPs, spectral filters, encoder, and decoder. These weights of SFNO are optimized during training.

The input to SFNO consists of seventy-four channels comprising the meteorological state at a given time. (Table 2). The model then predicts those same seventy-four channels at a future time of six hours, which also determines in the time step of the SFNO-BVMC ensemble future. In addition to the prognostic channels, we add three extra input channels: the cosine of the solar zenith angle, orography, and land-sea mask.

The existing implementation of SFNO from Bonev et al. (2023) makes forecasts for seventy-three total prognostic variables. channels.

In this study, we add <u>ERA5</u> 2-meter (2m) dewpoint temperature as another variable; for our SFNO training dataset, we obtain the 2m dewpoint temperature field from ERA5. Together, 2m dewpoint temperature and 2m air temperature provide an estimate of heat and humidity at the surface. Since we have trained SFNOs to predict both these variables, we can <u>assesssimulate</u> LLHI heat-humidity events. It is vital to assess the combination of both heat and humidity to characterize heat stress and LLHIs in a warming world (Vargas Zeppetello et al., 2022). <u>Olivetti and Messori (2024) evaluate deterministic While some</u> ML-based extreme weather forecasts, but they use <u>models have</u> 1000-hPa specific humidity as a proxy for surface humidity. They, Pasche et al. (2024) note that this approximation has limitations in predicting the surface heat stress and heat

index. We build on their Therefore, we add 2m dewpoint to more directly characterize moisture near the surface. In future work-by creating a model that predicts 2m temperature and dewpoint; also, while they analyze deterministic forecasts, our diagnostics pipeline is designed for extreme weather forecasts from ensembles. The, the addition of 2m dewpoint also enables could enable estimating Convective Available Potential Energy in the forecasts from SFNO. By quantifying the buildup of convective instability, this variable is useful for studying convective storms and thunderstorms. In total, we train SFNO to predict 74 meteorological variables, which are listed in Table 2.

We choose SFNO because its spherical design is well-suited for problems in earth science, and the The SFNO architecture includes scalable model parallel implementations, in which the model is split across multiple GPUs during training (Bonev et al., 2023; Kurth et al., 2023). Since SFNO can be split across multiple GPUs during training, we can We train large SFNOs and assess the effect of the SFNO size on the ensemble dispersion.

\_SFNO contains a number of hyperparameters that determine the total size of the model and its ensemble performance. Two such hyperparameters are the scale factor and the embedding dimension. The scale factor controls the level of spectral downsampling of specifies how much the input field, is spectrally downsampled when creating the latent representation. With larger downsampling, more aggressive downsampling, SFNO internally represents the input atmospheric state with reduced resolution. We speculate that this may reduce the effective resolution of the SFNO decreases, and finer-predictions (Brenowitz et al., 2024). With a reduced effective resolution, small-scale perturbations are blurred out. These perturbations would not grow appreciably during the model rollout, so the model spread and propagate upscale. Instead, they would be blurred out, and they would not cover the range of future weather states. Thus, we expect a model with a lower scale factor (less downsampling) to have larger result in increased spread among ensemble spreadmembers. The embedding dimension determines the size of the learned representation of the input fields (Pathak et al., 2022). A larger embedding dimension increases the number of learnable parameters in the SFNO, thereby requiring more GPU memory.

We compare three combinations of these hyperparameters: a *small* SFNO, a *medium* SFNO, and a *large* SFNO. The small SFNO has a scale factor 6 and embedding dimension of 220, the medium-sized model has a scale factor of 4 and embedding 850 hPa Temperature Lagged Ensemble

Figure 2. Comparing different versions of SFNO. (a) The 850 hPa temperature spread-error ratios are compared for lagged ensembles. A lagged ensemble is created by using nine adjacent time steps as initial conditions, and the spread-error is shown for each SFNO configurations. (b) Relative power spectra at a lead time of 360 hours (colored lines) for 850 hPa temperatures for a large SFNO (with a scale factor of 2 and an embed dimension of 620), a medium-sized SFNO (scale factor 4 and embed dimension 384), and a small SFNO (scale factor 6 and embed dimension 220). Spectra are computed relative to the ERA5 spectrum (horizontal black line).

-dimension of 384, and the large model has a scale factor of 2 and embedding dimension of 620. The small, medium, and large SFNOs have 48 million learned weights, 218 million learned weights, and 1.1 billion learned weights, respectively. Based on the number of weights, the large SFNOs are among the largest ML-based weather forecasting models currently available.

To select an SFNO architecture, we assess how these hyperparameters affect lagged ensemble spread-error ratio and spectral degradation. A lagged ensemble creates an ensemble is created by using nine adjacent time steps as initial conditions (Brankovic et al., 1990). Brenowitz et al. (2024) analyze the spread-error ratio of lagged ensembles to assess the intrinsic dispersion of deterministic ML weather models to assess their intrinsic dispersiveness. A spread-error ratio can be calculated from this ensemble. Ordinarily, benchmarking the ensemble performance would require generating and tuning thea full set of ensemble parameters (e.g. amplitude of perturbations, number of checkpoints, form of perturbations) separately for each architecture. This process is time-consuming, memory-intensive, and computationally demanding. Lagged ensembles readily enable comparison of different deterministic architectures without separately with minimal tuning ensembling methods for each architecture.parameters. In Figure 2a, the lagged ensemble spread-error ratio for 850 hPa temperature is highest closest to 1 for the large model, indicating that this model is best-suited for ensemble forecasting. The spread-error ratio systematically improves for the larger models. Brenowitz et al. (2024) find complementary results; they show that smaller scale factors favorably enhance dispersion-improve the spread-error ratio. Here, we consider the combined effect of changing both scale factor and embedding dimension.

We compare assess the extent to which the small, medium, and large SFNOs' spectra, and we assess the extent to which theySFNOs fully resolve the spectrum of the underlying ERA5 training data. A known problem with deterministic ML models

is that the small wavelengths are blurry (Kochkov et al., 2023). We attempt to suppress this blurring as much as possible by using a small scale factor and a large embedding dimension. In addition, we intentionally avoid using autoregressive training (Lam et al., 2023; Pathak et al., 2022; Keisler, 2022). In this method (sometimes called "multistep finetuning" or "multistep loss"), the ML model weights are optimized over multiple timesteps, not just a 1-step prediction. The goal of this method is to improve the forecast performance by training the ML model to perform well when autoregressively rolled out with its own predictions. Brenowitz et al. (2024) and Lang et al. (2024) hypothesize that autoregressive training could contribute to spectral degradation. This method may effectively increase the time step of the model, making it more similar to an ensemble mean (Lang et al., 2024). Many deterministic models' initial 1-step forecasts are blurry, and with this method, their forecasts get increasingly blurry with lead time. Because we do not use autoregressive finetuning, we hypothesize that SFNO has spectra that stay constant with lead time (see Section 3.2 for more discussion).

With this design decision, we also reduce the computational requirements of training SFNO. Autoregressive fine-tuning requires significant GPU memory and computation time because it calculates gradients across multiple model steps. With these computational savings, we train large SFNOs with a small scale factor and a large embedding dimension. These design choices allow our configuration of SFNO to hold as much high-resolution information in its internal representation as possible. Figure 2b shows that the larger models (with lower scale factors and larger embed dimension) have less spectral degradation and are better able to preserve the spectra of ERA5. Based on these two tests Figure 2a and b, we useselect the large version of SFNO, with a scale factor as our final set of 2 and embed dimension of 620 hyperparameters. This version of SFNO trains in 16 hours on 256 80GB NVIDIA A100 GPUs. It leverages data parallelism, in which the batch size of 64 is split up across different GPUs, and spatial model parallelism, in which the input field is divided into four latitude bands. These four bands are split across four GPUs (one GPU per section), and the SFNO architecture is distributed to train with spatial model parallelism. Each SFNO checkpoint trains for 70 epochs using a pressure-weighted mean squared error loss function (Lam et al., 2023).

We note that there are many possible combinations of the scale factor, embedding dimension, and other training hyperparameters. We do not conduct comprehensive hyperparameter tuning via a grid search. Such an experiment would be very computationally expensive, due to the large number of hyperparameter combinations. Instead, we optimize the scale factor and embedding dimension because of their direct relevance to spectral degradation. Instead of Rather than hyperparameter tuning, we choose to expend our compute budget on training as many checkpoints as possible. With multiple checkpoints, we aim to try to span the model space of all possible SFNO checkpoints that have our chosen architecture and hyperparameters. By having as . With many SFNO checkpoints as possible, we hope to increase our coverage of extreme weather events with a thorough representation of model uncertainty.

#### 2.2 Selecting a number of checkpoints for the ensemble

We train 34 SFNO checkpoints from scratch, at which point we determined determine that 29 checkpoints adequately sample the ensemble spread as described below. We consider experiment with using different numbers of checkpoints in the size of the ensemble, from 4 checkpoints to 34 checkpoints, at intervals of 5 checkpoints. For each ensemble size, we conduct 100

bootstrap samples with replacement from the 34 checkpoints. Figure 3 shows the ensemble-resulting ensemble spread obtained from these bootstrap samples. The ensemble spread is calculated as the global mean ensemble variance at each grid cell; it is calculated for a 120-hour lead time and averaged over forecasts initialized at 52 initialization dates (one initialization per week of 2018). We choose 120 hours because this timescale allows for synoptic-scale errors to grow, and given its importance for weather forecasting, we hope to represent model uncertainty for this time period as accurately as possible. Figure 3 shows that the ensemble spread asymptotes at approximately twenty-nine checkpoints. We conclude that twenty-nine checkpoints adequately sample the underlying population of all possible SFNO checkpoints with our selection of hyperparameters. In our ensemble results for the remainder of this paper, we use twenty-nine checkpoints. We open-source all

Figure 3. Ensemble spread from different numbers of checkpoints. Ensemble spread is calculated as the square root of time-mean, globalmean variance (Fortin et al., 2014). A correction factor of N-1 is applied to account for different ensemble sizes in the unbiased estimator of variance. At a lead time of five days, ensemble spread is averaged over forecasts from fifty-two initial conditions in the validation set (one per week starting 01-02-2018). Ensemble spread is shown for total column water vapor (left), 10m wind speed (middle), and 2m temperature (right). For each number of SFNO checkpoints, 200 estimates of ensemble spread are obtained by taking 100 bootstrap random samples of the SFNO checkpoints. The box-and-whiskers visualize the distribution of these 200 trials: the middle of the box is the median, the ends of the box are the first and third quartile of the data, and the ends of the box are correspond to the minimum and maximum.

34 model checkpoints (each with 1.1 billion learned weights) as a resource to the community, to explore further the benefit of multiple SFNOs on forecasting atmospheric phenomena.

#### 2.3 Bred vectors with SFNO

Bred vectors are a computationally efficient way to sample the fastest growing modes of the atmosphere (Toth and Kalnay, 1993). In Figure 4, we generate bred vectors using the following methodology:

1. Generate spherical random noise correlated on 500 km length scales. Add this noise as a perturbation to 500 hPa geopotential at time  $t_{-32}$ 

- forecast using(with the perturbed input) and a control forecast using(with the unperturbed input<sub>-</sub>).
- 2.3. Subtract the control forecast from the perturbed forecast. Use this difference as the perturbation. (Unlike the initial noise in Step 1, this perturbation is applied to all variables and pressure levels, not just Z500.)
- 3.4. Rescale the perturbation in each hemisphere to the target amplitude of the perturbation.
- **4.5**. Repeat steps (2)-(4) for  $t_{-2}t_{-1}$  and  $t_0$ .

The resulting perturbation is added to or subtracted from  $t_0$ . Using this perturbed initial condition, SFNO generates a 360hour forecast, which serves a perturbed member in the ensemble.

The amplitude of the bred vectors is determined by the deterministic RMSE of SFNO at 48 hours, multiplied by a scaling factor of 0.35. At early lead times, the deterministic and ensemble mean RMSE of an ensemble forecast are similar. This factor is a tuning parameter. Since this parameter is less than 1, it reduces the perturbation amplitude. At early lead times, the deterministic and ensemble mean RMSE of an ensemble forecast are similar. To satisfy criteria for

Figure 4. Diagram of generating bred vectors. This diagram details the process of generating bred vectors used for developing initial condition perturbations at  $t_0$ . First, using the input three time steps before  $t_0$  (denoted  $t_{-2}$ ), random noise is added to 500 hPa geopotential (z500). This noise respects spherical geometry and has a spatial correlation length scale of 500 km. With  $t_{-2}$  as the initial condition, the perturbed forecast is subtracted from the control forecast. This difference is rescaled and used as a new perturbation, which is added to  $t_{-1}$ . This process is repeated for  $t_0$ . For each variable during every step of the breeding process, the amplitude of the perturbation is scaled to be 0.35 \* the deterministic RMSE of SFNO at 48 hours

÷

-statistical exchangeability, the ensemble spread should match its ensemble mean RMSE. Thus, weWe use the deterministic RMSE (with a tuning parameter) as a proxy for the desired spread level at early lead times. This approach provides a clear guide for the amplitude of each variable at each pressure level. TuningManually tuning these amplitudes across variables and levels would be challenging, since there are seventy-four different input variables. Figure A1B1 shows the actual amplitude for each of the seventy-four variables.

We adopt two design choices from Toth and Kalnay (1997) and Toth and Kalnay (1993): centered perturbations and hemispheric-dependent amplitudes. For each learned bred vector, we both add it to and subtract it to initial condition; this creates two separate perturbations (one positive and one negative). Centered perturbations improved the performance of the ensemble mean RMSE on the 2018 validation set (not shown). Additionally, we rescale the amplitudes separately for the Northern Hemisphere extratropics and the Southern Hemisphere extratropics. To prevent jump discontinuities in the perturbation amplitudes near 20° N and 20° S, a linearly interpolated rescaling factor is used in the tropics. Hemispheric rescaling prevents one hemisphere from dominating the perturbation amplitude. All perturbations are clipped to ensure that total column water vapor and specific humidity cannot be negative. See Appendix Section E for a note about our implementation of bred vectors.

In Step 2 of Figure 4, we add correlated spherical noise to 500 hPa geopotential (Z500). The noise has a correlation length scale of 500 km, and it has the same structure as noise of the Stochastic Perturbed Parameterized Tendency scheme used at ECMWF (Leutbecher and Palmer, 2008). We only add the initial noise to Z500, to avoid perturbing different fields in opposing and possibly contradictory directions. For instance, positively perturbing total column water vapor but negatively perturbing specific humidity at 1000, 925, and 850 hPaon the lower pressure levels would likely be unphysical. Z500 is a natural choice of initial field to perturb because it is the steering flow in the extratropics. Because Since it is a smooth field on an isobaric surface, correlated spherical noise is an appropriately structured additive perturbation. On the other hand, correlated spherical noise would not serve well as an additive perturbation to surface fields, which have sharp discontinuities due to orography and land sea contrasts. We design the bred vectors with the goal of keeping the perturbed input as close to the training dataset as possible. We minimize the extent

.

of directly prescribed perturbations, and the majority of the perturbation structure is generated from the breeding process with SFNO itself. To start the breeding cycle, the initial perturbation is applied to Z500, but for all subsequent cycles, all 74 input variables are perturbed. In this manner, we develop a mutually consistent way of perturbing all input channels.

We test our bred vectors by evaluating spread-error performance on the validation year: 2018. Figure 5 visualizes sample bred vectors for various input fields and channels. These perturbations contain some desirable qualities. First, they contain a land-sea contrast for surface fields such as 10m wind speed and 2m temperature. For these surface fields, perturbations have distinct amplitudes and spatial scales in this example, the 2m temperature perturbation has an amplitude of 0.56 K over the

land and <u>0.27 K over the ocean</u>, and the 10m wind speed perturbation has an amplitude of 0.45 m/s over land and 0.66 m/s over the ocean. The specific humidity perturbations are stronger in the tropics than at the poles, in line with a strongthe equator-to-pole moisture gradient. This is These physical qualities of bred vectors are a benefit of using bred vectors, compared to perturbing multiple input variables simply with over simple spherical noise, as in GraphCast-Perturbed (Price et al., 2023), or Perlin noise (Bi et al., 2023).

We initially presented bred vectors and multiple checkpoints in Collins et al. (2024). Concurrently, Baño-Medina et al. (2024) also released a preprint using bred vectors and multiple trained models. The results in Baño-Medina et al. (2024) serve as excellent independent validation of bred vectors and multiple checkpoints. They validate their method from Jan 10 to Feb 28 (with 50 forecast initial dates), and they show promising results, particularly at certain latitudes and land regions. We comprehensively show that SFNO-BVMC is competitive with IFS on global mean quantities using forecasts from a full year (732 forecast initial dates for 2020 and 92 for summer 2023). We further validate our ensemble with a unique pipeline for extreme diagnostics and spectral diagnostics of each ensemble member and the ensemble mean. While their method uses Adaptive Fourier Neural Operators (AFNO) (Pathak et al., 2022), we use SFNO, a successor to AFNO that is more stable and has better skill. We train all 29 SFNOs from scratch, whereas they sample multiple models from 3 training runs. To compare methodologies with Figure 2 in Baño-Medina et al. (2024), we present a diagram of how we generated bred vectors in SFNOBVMCSFNO-BVMC. The boxed quantities in Figure 4 represent the unique methodological details of our approach. We add spherical initial noise to Z500 (compared to Gaussian noise), start the breeding cycle 3 timesteps before the initialization date (compared to Jan 1, 2018), and

Figure 5. Sample visualizations of the learned bred vectors. For a sample initial time (June 18, 2020 00:00 UTC), the bred vectors are visualized for six different input fields: 850 hPa specific humidity, 10m wind speed, surface pressure, 2m temperature, 500 hPa geopotential, and 850 hPa temperature.

-use the deterministic RMSE as the bred vector amplitude. In Part II, we assess the forecasts from bred vectors and multiple checkpoints at scale, with a significantly larger ensemble- than in Baño-Medina et al. (2024).

#### 2.4 Contributions of bred vectors and multiple checkpoints to the ensemble calibration

In SFNO-BVMC, the bred vectors and multiple trained model checkpoints both contribute to ensemble spread and calibration. Bred vectors are a flow-dependent initial condition perturbation: they are calculated independently for each checkpoint, and they use the preceding threetwo time steps to generate the perturbation according tobased on the current flow in the atmosphere. At longer lead times, when there is less dependence on the initial conditions, multi-checkpointing causes the spread-error ratio to approach 1; this is consistent with our expectations from the ensemble in Weyn et al. (2021). In Figure 6, we show the spread-error ratios from three different ensembles: Figure 6a has only 29 checkpoints and no bred vectors, Figure 6b has 1 checkpoint and 29 bred vectors (each added to and subtracted the initial condition), and Figure 6c has 29 checkpoints and 1 bred vector (added to and subtracted from the initial condition). Figure 6a has 29 ensemble members, while Figures 6b and c have 58-member ensembles. As a model perturbation, multi-checkpointing does not represent the uncertainty arising from an imperfect initial condition. Therefore, the multi-checkpoint ensemble is underdispersive at early lead times. On the other hand, the ensemble composed only of bred vectors is underdispersive on synoptic time scales (3-5 days) when representing model uncertainty also becomes important for obtaining good calibration grand ensemble in Weyn et al. (2021). In Figure 6, we show

# 3 Ensemble Diagnostics

<u>Ultimately, with SFNO-BVMC</u>, we hope to study <u>LLHIs</u>. This requires a calibrated ensemble with reliable probabilistic forecasts. <u>SFNO-BVMC</u> is a novel way to create ensemble forecasts from deterministic ML models. Therefore, in the following section, we present a diagnostics pipeline to evaluate the SFNO-BVMC ensemble and compare it to the IFS ensemble. <u>We</u>

Figure 6. Contributions of bred vectors and multiple checkpoints to spread-error relations. (a) shows the spread-error relation obtained from an ensemble only composed of multiple checkpoints. This ensemble has twenty-nine members, one for each checkpoint. (b) shows

the same for an ensemble of fifty-eight members, using only bred vectors for initial condition perturbations. (c) shows the spread-error relation for an ensemble composed of fifty-eight members, with one bred vector added and subtracted from the initial condition for each model checkpoint. Spread-error ratios are averaged across fifty-two initial conditions, one per week starting 01-02-2018, in 2018. Successful ensemble forecasts have a spread-error ratio of 1.

the spread-error ratios from three different ensembles: Figure 6a has only 29 checkpoints and no bred vectors, Figure 6b has 1 checkpoint and 29 bred vectors (each added to and subtracted the initial condition), and Figure 6c has 29 checkpoints and 1 bred vector (added to and subtracted from the initial condition). Figure 6a has 29 ensemble members, while Figure 6s b and c have 58 members ensembles. As a model perturbation, multi-checkpointing does not represent the uncertainty arising from an imperfect initial condition. Therefore, the multi-checkpoint ensemble is underdispersive at early lead times. On the other hand, the ensemble composed only of bred vectors is underdispersive on synoptic time scales (3-5 days) when representing model uncertainty also becomes important for obtaining good calibration (Palmer, 2018).

#### 3 Ensemble Diagnostics

Ultimately, with SFNO BVMC, we hope to analyze the statistics of LLHIs. This requires a calibrated ensemble with reliable probabilistic forecasts. SFNO BVMC is a novel way to create ensemble forecasts from deterministic ML models. Therefore, in the following section, we present a diagnostics pipeline to evaluate the SFNO-BVMC ensemble and compare it to the IFS ensemble. We first evaluate SFNO-BVMC using diagnostics that evaluate overall performance. Next, we assess SFNO-BVMC's control, perturbed, and ensemble mean spectra. Finally, we present diagnostics specifically focused on extreme weather forecasts. We open-source the code for these diagnostics (see Data Availability section), and we hope that it can be used to guide future ML model development. For a fair comparison for all diagnostics, we validate IFS against ECMWF's operational analysis and SFNO-BVMC against ERA5. IFS is initialized with this operational analysis, not the ERA5 reanalysis, so it has a different verification dataset. All diagnostics show SFNO-BVMC resultsscores with 58 members, and IFS ENS resultsscores with 50 members. SFNO-BVMC has 58 members: 29 checkpoints and 1 bred vector per checkpoint (added to and subtracted from the initial condition). Because of the use of 29 checkpoints and centered bred vector perturbations, SFNO-BVMC cannot be evaluated with an ensemble size smaller than fifty-eight members. However, while While there are unbiased-versions of the metrics that are corrected for ensemble size, the difference in the metrics due to different ensemble size would be sufficiently small that the diagnostics still allow for fair evaluation comparison between the 50-member IFS and 58-member SFNO-BVMC.

#### 3.1 Mean Diagnostics

We validate the overall quality of the ensemble on three diagnostics: continuous ranked probability score (CRPS), spreaderror ratio, and ensemble mean RMSE. First, CRPS evaluates a probabilistic forecast of a ground truth value. It is a summary score of the performance of the ensemble forecast. The formula for CPRS at a given grid cell is

Figure 7. CRPS of SFNO-BVMC and IFS ENS. SFNO-BVMC is a 58 member ensemble that uses 29 SFNO checkpoints trained from scratch, and two initial condition perturbations per checkpoint. The two initial condition perturbations come from a single bred vector that is added to and subtracted from the initial condition. Scores are calculated over 732 initial conditions (two per day at 00 UTC and 12 UTC) for 2020, which is the test set year. SFNO-BVMC is validated against ERA5, and IFS ENS is validated against ECMWF's operational analysis. IFS ENS scores are taken from WeatherBench 2 Rasp et al. (2024).

$$(F,y) = \int_{-\infty}^{\infty} (F(z) - 1\{y \le z\})^2 dz \qquad \text{CRPS}$$

$$= E_F |X - y| - \frac{1}{2} E_F |X - X'| \qquad (1)$$

where X and X' are random variables drawn from the cumulative distribution function (CDF) of the ensemble forecast F. Here, y is the verification value (ERA5 for SFNO-BVMC and operational analysis for IFS ENS).

Figure 7 compares the global mean CRPS of SFNO-BVMC to that of IFS ENS on five different variables. On 850 hPa temperature, 2m temperature, 850 hPa specific humidity, and 500 hPa geopotential, SFNO-BVMC lags approximately 12–18 hours behind IFS ENS, though their performance is comparable. SFNO-BVMC does match IFS ENS on the 10m zonal (u component) wind.

Second, an essential requirement for an ensemble weather forecast is that the ensemble spread must match its skill (Fortin et al., 2014); the spread-error ratio should be 1. This result is derived statistically based on the idea of exchangeability between ensemble members: each ensemble member should be statistically indistinguishable from each other and from the forecasts (Fortin et al., 2014; Palmer et al., 2006). The spread is the square root of the global-mean ensemble variance. Similarly, the error is the square root of the global-mean ensemble MSE. See Section BC for a detailed description of calculating the spread and error across multiple forecasts initialized on different initial times. Figure 8 demonstrates that SFNO-BVMC obtains spread-error

Figure 8. Spread-Error Ratio of SFNO-BVMC and IFS ENS. SFNO-BVMC is the same 58 member ensemble described in Figure 7. Spread-error ratios are calculated over 732 initial conditions (two per day at 00 UTC and 12 UTC) for 2020. SFNO-BVMC is validated against ERA5, and IFS ENS is validated against ECMWF's operational analysis. IFS ENS scores are taken from WeatherBench 2 Rasp et al.- (2024).

ratios that approach 1, and it has comparable performance to IFS ENS. At early lead times, SFNO-BVMC is underdispersive for all variables except Z500, but the spread skill ratio approaches 1 for longer lead times.

Finally, we evaluate the ensemble mean RMSE of SFNO-BVMC and IFS ENS (Figure 9). Their scores are comparable, with SFNO-BVMC lagging close behind the IFS ensemble mean, and both models have an ensemble mean RMSE that converges to climatology at 360 hours (14 days).

On these aggregate metrics, SFNO-BVMC is often eighteen hours behind IFS ENS, so its performance is slightly worse but still comparable. Through large SFNOs with a high-resolution, expressive internal state, bred vectors, and multi-checkpointing, this ensemble has significantly improved calibration, compared to previous work using lagged ensembles (Brenowitz et al., 2024). It serves as a benchmark for the calibration potential for deterministic ML models, and it can be compared to recent models which optimize for an ensemble objective. While IFS ENS has been an established weather forecasting model for decades, SFNO is still a new architecture. Improving the skill of the SFNO architecture itself is an important area of future research. However, in this manuscript, our main goal is not primarily to create the most skillful weather forecasting model;

rather, we hope to explore huge ensembles and low-likelihood events at the tail of the ensemble forecast distribution. SFNO-BVMC is orders of magnitude less computationally expensive than IFS, so it uniquely enables the creation of huge ensembles of forecasts. These allow for unprecedented sampling of internal variability and an analysis of extreme statistics, as presented in Part II of this paper.

#### 3.2 Spectral Diagnostics

A common issue with deterministic machine learning weather models is that their forecasts tend to be "blurry" (Kochkov et al., 2023). As a metric to measure and quantify this blurriness, existing work compares the spectra of the ML predictions to the spectra of ERA5. The spectral analyses show that ML models have reduced power at small wavelengths compared to ERA5. Deterministic ML models are often trained using the MSE loss function, which strongly penalizes sharp forecasts in

Figure 9. Ensemble Mean RMSE of SFNO-BVMC and IFS ENS. SFNO-BVMC is the same 58 member ensemble described in Figure 7. Scores are calculated from forecasts initialized at 732 initial conditions (two per day at 00 UTC and 12 UTC) for 2020. SFNO-BVMC is validated against ERA5, and IFS ENS is validated against ECMWF's operational analysis. IFS ENS scores are taken from WeatherBench 2 Rasp et al. (2024).

the wrong place variability and an analysis of extreme statistics, as presented in Part II of this paper. Additionally, while IFS ENS has been an established weather forecasting model for decades, SFNO is still a new architecture. Improving the skill of the SFNO architecture itself is an important area of future research.

#### 3.2 Spectral Diagnostics

A known problem with deterministic ML weather models is that their forecasts are "blurry" (Kochkov et al., 2023). Compared to ERA5, they have reduced power at small wavelengths. Deterministic ML models are often trained using the MSE loss function, which strongly penalizes sharp forecasts in the wrong place. This is referred to as the double penalty problem

(Mittermaier, 2014), in which an ensemble is penalized once for predicting a storm in the wrong place and another time for missing the correct location of the storm. To avoid the double penalty from the mean squared error, ML models may learn to predict smooth, blurred solutions. Existing work has noted that these smooth ML predictions that appear closer to an ensemble mean (Agrawal et al., 2023; Brenowitz et al., 2024), rather than an individual ensemble member.

However, a key feature of the Regarding spectral performance, there are two desirable characteristics. These characteristics distinguish an individual ensemble member from an ensemble mean:

- 1. During the rollout, it is not just that preferable for the forecasts are blurry. The spectra must increasingly blur of each ensemble member to stay constant with lead time. With this characteristic, each ensemble member maintains a realistic representation of the atmospheric state during the rollout.
- 2. During the rollout, it is preferable for the spectra of the ensemble mean to realistically degrade with lead time (Bonavita, 2023). As the ensemble members spread more and their trajectories diverge during the forecast rollout, the ensemble mean should become blurrier, and its spectra should increasingly degrade at small wavelengths. In particular, on synoptic time scales (around 3-5 days), when error growth becomes nonlinear, the IFS ensemble mean displays a sharp decline in power around 1000 kilometer wavelengths (Bonavita, 2023).

(Bonavita, 2023). This introduces two important criteria for an ML-based weather forecast: whether the forecasts are blurry, and whether they get increasingly blurry with lead time.

On the first criterion, SFNO-BVMC ensemble members do contain some blurriness, like many other deterministic ML models. On the second criterion, crucially, their characteristic, SFNO-BVMC spectra remain constant through the 360-hour rollout (Figure 10 and Figure 11). This contrasts with GraphCast and AIFS; those deterministic ML models do increasingly blur with lead time (Kochkov et al., 2023; Lang et al., 2024). WeBrenowitz et al. (2024) and Lang et al. (2024) hypothesize that autoregressive fine tuning could be responsible for this behavior. In autoregressive fine tuning, the ML model weights are optimized over multiple timesteps. Normally, during training, the model weights are optimized to minimize the MSE of just 1 timestep of the forecast, but during the autoregressive fine tuning phase, the weights are optimized based on the predictions and ground truth after multiple time steps. The goal of this method is to improve the performance during rollout. Autoregressive fine tuning may effectively increase the time step of the model, making it more similar to an ensemble mean (Lang et al., 2024) and contributing to increased blurring during the rollout. To minimize this spectral degradation, we do not conduct any autoregressive fine tuning. The SFNO trained here is only trained to predict six hours ahead, and its autoregressive performance is not optimized. With this design decision, we reduce the training computational requirements, as autoregressive fine tuning is intensive in GPU memory and computation time. We use the savings from this choice to train an SFNO with a small scale factor and large embedding dimension. SFNO-BVMC These design choices allow our configuration of SFNO to hold as much high resolution information in its internal representation as possible.

While the control and perturbed spectra remain constant because of our intentional choice not to use autoregressive training.

Through this test, we validate that the individual members' predictions do not collapse into the ensemble mean. This is a crucial test of the physical fidelity of SFNO-BVMC. Because each SFNO-BVMC ensemble member's spectrum is constant through the rollout, the ensemble members maintain their ability to through the rollout, the SFNO-BVMC ensemble mean does increasingly blur with lead time.resolve extreme weather. If their spectra degraded with lead time, then the forecasts may become too blurry to predict highly localized extreme events. At a lead time of 360 hours, the perturbed members maintain similar spectra as the control member (Figure 11), and at the initial time, they have similar spectral characteristics as the unperturbed ERA5 initial condition (Figure D5).

An important caveat is that even though the spectra are constant during the rollout, they are still somewhat degraded compared to ERA5 (Figure 2b) We have not solved the problem of blurry forecasts entirely. We have minimized it as much as possible by using a large embedding dimension and a small scale factor, which increase the resolution of the latent representation of the input, and by intentionally avoiding multistep finetuning. However, our deterministic training setup still results in blurring with the use of the MSE loss function and large six-hour timesteps, and alleviating this problem is an important avenue for future research.

On the second characteristic, the SFNO-BVMC ensemble mean realistically degrades with lead time: it has a similar ensemble mean spectra as the IFS ensemble mean. Figure 12 shows that the ensemble means of SFNO-BVMC and IFS ENS similarly degrade in power after 24 hours, 120 hours, and 240 hours. For Z500, there is a notable decline in power between lead times of 24 hours and 120 hours. This sharp decline is due to the nonlinear error growth that characterizes forecasts at lead times of 3–5 days. On synoptic Scales (~1000 km in space and 3–5 days in time), SFNO-BVMC's ensemble mean has a similar decline in power as IFS ENS. This increases our trust that the ensemble members trajectories realistically diverge, and the ensemble is correctly representing synoptic error growth.

These two results pass a crucial test laid out by Bonavita (2023). They originally posed this test comparing the spectra of a deterministic PanGu ML model and the IFS ensemble mean. Despite the blurring in PanGu, they show that a control run of PanGu does not successfully mimic the IFS ensemble mean spectrum. In our work, we have created an ensemble prediction system from multiple deterministic ML models that meets the above two characteristics.

# 3.3 Extreme Diagnostics

The preceding analysis has evaluated ensemble weather forecasts from SFNO-BVMC on overall weather. This is necessary but as yet insufficient validation for our main scientific interest in LLHIs. Since extreme weather events are rare in space and time, they contribute relatively little to these scores. Hereafter, we focus on diagnostics specifically designed to validate the performance of SFNO-BVMC on extreme weather. We complement these diagnostics with a case study of the Phoenix 2023 heatwave in Figure A1.

#### 3.3.1 Extreme Forecast Index

As part of its IFS evaluation, ECMWF releases a Supplemental Score on Extremes (Haiden et al., 2023). This score is based on the Extreme Forecast Index (EFI). Using an ensemble forecast and its associated model climatology, the EFI is a unitless

Figure 10. Control Spectra. Spectra from the control member of SFNO-BVMC averaged across forecasts from fifty-two initial times, one per week starting January 2, 2020. Spectra are shown for 850 hPa temperature, 2m temperature, and 500 hPa geopotential. Note the different scales on the y-axis for each variable.

Figure 11. Perturbed Spectra. Spectra of the control member and each perturbed member from a 58-member SFNO-BVMC ensemble are shown. The shading denotes the range of all the perturbed members. Spectra are averaged across forecasts from fifty-two initial times, one per week starting January 2, 2020.

scales (~1000 km in space and 3–5 days in time), SFNO-BVMC's ensemble mean has a similar decline in power as IFS ENS.

This increases our trust that the ensemble members trajectories realistically diverge, and the ensemble is faithfully representing synoptic error growth. Bonavita (2023) originally posed this test comparing the spectra of a deterministic PanGu

ML model and the IFS ensemble mean. Despite the blurring in PanGu, they show that a control run of PanGu does not successfully mimic the IFS ensemble mean spectrum.

Through these spectral diagnostics, we validate that the individual members' predictions do not collapse into the ensemble mean. This is a crucial test of the physical fidelity of SFNO-BVMC. Because each SFNO-BVMC ensemble member's spectrum is constant through the rollout, the ensemble members maintain their ability to predict extreme weather. If their spectra degraded with lead time, then the forecasts would become too blurry to predict localized extreme events.

#### 3.3 Extreme Diagnostics

The preceding analysis has evaluated ensemble weather forecasts from SFNO-BVMC on overall weather. This is necessary but as yet insufficient validation for our main scientific interest in LLHIs. Since extreme weather events are rare in space and time, they contribute relatively little to these scores. Hereafter, we focus on diagnostics specifically designed to validate the performance of SFNO-BVMC on extreme weather.

#### 3.3.1 Extreme Forecast Index

As part of its IFS evaluation, ECMWF releases a Supplemental Score on Extremes (Haiden et al., 2023). This score is based on the Extreme Forecast Index (EFI). Using an ensemble forecast and its associated model climatology, the EFI is a unitless quantity that quantifies how unusual an ensemble forecast is. The EFI ranges from -1 (unusually cold) to 1 (unusually hot). The EFI measures the distance between the ensemble forecast CDF and the model climatology CDF (Lalaurette, 2002; Zsótér, 2006). The formula for the EFI is

The formula for the EFI is

Figure 12. Ensemble Mean Spectra. The spectra of the ensemble mean of SFNO-BVMC and IFS ENS are shown. Spectra are averaged across forecasts from fifty-two initial times, one per week starting January 2, 2020. Spectra are shown for 850 hPa temperature (left) and 500 hPa geopotential (right).

EFI (2)

where Q is a percentile, and Qf(Q) denotes the proportion of ensemble members lying below the Q percentile calculated from the model climatology. The model climatology is calculated for each lead time for each grid cell.

To calculate the EFI, a model climatology is necessary. The model climatology encapsulates the expected weather for a given time of year. For a given initial day, ECMWF creates a model climatology (called M-Climate) using hindcasts from 9 initial dates per year, 20 years, and 11 ensemble members (for a total of 1980 values). The CDF of these 1980 values represents the model climatology. This CDF is defined at each grid cell for each lead time, and it is used to calculate the Qf(Q) term in Equation 2. See Lavers et al. (2016) for more information on the M-Climate definition.

We generate a model climatology of SFNO-BVMC using the same parameters as ECMWF's M-Climate, except the SFNOBVMC M-Climate uses 12 ensemble members, not 11. This is due to the use of centered (positive and negative) bred vector perturbations, which requires an even number of ensemble members. After creating the climatology of SFNO-BVMC, we calculate the CDF of the model climate for each lead time for each grid cell. We use these CDFs to calculate the EFI for the SFNO-BVMC forecasts initialized on each day of summer 2023. Figure 13 visualizes a sample EFI from SFNO-BVMC and

\_IFS four days into a forecast on an arbitrary summer day. The IFS EFI values are directly downloaded from the ECMWF MARS-data server. The SFNO-BVMC and IFS EFI values have excellent agreement across the globe (Figure 13). Notable features include pronounced heatwaves over much of Africa, South America, and the Midwest of the United States. The strong El Niño pattern in the tropical Pacific appears in the EFI for both SFNO-BVMC and IFS ENS. Visually, SFNO-BVMC has a smoother EFI than IFS ENS. This is a consequence of the blurriness of the SFNO 2m temperature predictions. Despite this, however, the SFNO EFI can still predict large-scale extremes, and the two models have similar scores on the extreme diagnostics below.

Figure 14 shows that IFS and SFNO-BVMC have highly correlated EFIs throughout summer 2023. Therefore, in principle, these two ensemble prediction systems offer comparable extreme forecasts and could be used to forecast various extreme events of interest. The EFI encapsulates the ability of each model to forecast extreme temperatures.

Therefore, in principle, the EFI

Figure 13. Visualization of the Extreme Forecast Index from SFNO-BVMC and IFS ENS. For each grid cell and lead time, the Extreme Forecast Index (EFI) is a unitless metric that represents the distance between the model climatology and the current ensemble forecast. It ranges from -1 (anomalously cold) to 1 (anomalously hot). For a sample 4-day forecast initialized on August 19, 2023, the EFI from the 58-member SFNO-BVMC is compared to the EFI from IFS ENS: the global latitude-weighted correlation is 0.89.

Figure 14. Comparing SFNO-BVMC and IFS ENS Extreme Forecast Index in boreal summer 2023. (a) shows the latitude-weighted spatial correlation between IFS ENS EFI and SFNO-BVMC EFI as a function of lead time. (b) shows the latitude-weighted 2D histogram between the SFNO-BVMC EFI and the IFS ENS EFI at a lead time of 5 days. Figures (a) and (b) are averaged using forecasts initialized over ninety-two initialization days, one per day (00 UTC) for each day in June, July, and August 2023.

Visually, SFNO-BVMC has a smoother EFI than IFS ENS. This is a consequence of the blurriness of the SFNO 2m temperature predictions. While the embedding dimension and scale factors mitigate this blurriness as much as possible, the SFNOBVMC model climatology and ensemble forecasts have this artifact. Therefore, the EFI values also appear blurry. Despite this,

however, the SFNO EFI can still predict large-scale extremes, and the two models have similar scores on the extreme diagnostics below.

Figure 14 shows that IFS and SFNO-BVMC have highly correlated EFIs throughout summer 2023. Therefore, in principle, these two ensemble prediction systems offer comparable extreme forecasts and could be used to forecast various extreme events of interest. The EFI encapsulates the ability of each model to forecast extreme temperatures. Therefore, in principle, the EFI similarity between SFNO-BVMC and IFS means that they have similarly skillful extreme weather forecasts, including heat extremes and cold extremes of varying severity.

The EFI itself does not measure the accuracy of a forecast; it only measures how extreme or unusual a forecast is by comparing a given forecast to the model climatology. To evaluate the accuracy of the extreme forecast, the EFI is compared

Figure 15. Extreme Diagnostics of SFNO-BVMC and IFS ENS. Diagnostics are averaged over forecasts initialized at 00 UTC for each day in June, July, and August 2023 (total of ninety-initialization days). SFNO-BVMC is validated against ERA5, and IFS ENS is validated against the ECMWF operational analysis. (a) measures the Receiver Operating Characteristic of the Area Under the Curve. Higher is better. (b) measures the threshold-weighted CRPS. Lower is better. (c) measures the reliability diagram, which compares the forecast probability to the observed occurrence. Reliable ensemble forecasts appear along the one-to-one line.

-to an observational dataset to assess if the extreme forecasts match observations. We follow ECMWF's validation strategy of using a Receiver Operating Characteristic curve to assess how well the EFI predicts the verification values. The ROC curve could be calculated to assess how well the EFI predicts extreme temperatures.

# 3.3.2 Reliability and Discrimination

Two key aspects of an ensemble forecast are its reliability and its discrimination. Measured by reliability diagrams, a forecast's reliability evaluates whether the predicted probability of extreme weather matches the observed occurrence. Measured by Receiver Operating Characteristic (ROC) curves, forecast discrimination is the ability to distinguish between an extreme

weather event and a not-extreme weather event. A ROC curve can be created for each forecast lead time, and it is summarized by the

ROC Area Under Curve (AUC) score. We calculated the ROC AUC for each lead time, and a purely random forecast would have an ROC AUC value of 0.5. A perfect forecast would have an ROC AUC value of 1.

Reliability diagrams and ROC curves are calculated by comparing two quantities: a binarized ground truth value (1 or 0, for extreme and not extreme) and a continuous ensemble forecast between 1 and 0.

A key validation criterion is the threshold defining extreme vs. not extreme. To enable future comparison with GenCast, weWe calculate our threshold for extreme temperature using the same definition as Price et al. (2023). Using the years 1992-2016 of ERA5, we calculate the climatological 95th percentile 2m temperature for each grid cell.

These percentiles are calculated for each time of day (00:00, 06:00, 12:00, and 18:00 UTC) for each month. This results in 48 different thresholds in total. This definition of extreme accounts for the diurnal and seasonal cycles: an event is considered extreme if it is hot for the time of day and time of year. It also thus includes warm nighttime temperatures, which has have important implications for fire (Balch et al., 2022) and human health (Murage et al., 2017; He et al., 2022), and warm winters, which has have important implications for agriculture (Lu et al., 2022). This is a different rationale than defining extreme weather using an absolute temperature threshold or a threshold based only on the summer daily maximum.

Figure 15c shows that SFNO-BVMC and IFS ENS are similarly reliable in their prediction of extreme warm\_2m temperatures at these lead times-of 120 and 240 hours. To create the reliability diagram in Figure 15a, the ground truth dataset is binarized using the extreme temperature threshold defined described above: one. The "Forecast Probability" is a continuous value from 0 to 1, indicating the proportion of the ensemble that exceeds the threshold. Over all grid cells and initial times of summer 2023, the reliability diagram compares the probabilistic forecasts of extreme events to their actual occurrence. In addition to the lead times in Figure 15c), we visualize the reliability diagrams for each month other lead times (Supplemental Figure D1) and variables. We show that SFNO-BVMC also performs reliably when forecasting the heat index at lead times of 48, 96, 120, and 240 hours. For 10m wind speed and cold extremes, SFNO-BVMC matches the performance of the IFS ensemble (Figure D2 and Figure D3). However, we also show that at 240 hour lead times, the model is not reliable when it confidently (greater than 50% chance) forecasts wind extremes or cold temperature extremes (see Appendix D and Figure D4 for each time of day. The ensemble prediction more discussion). This is an area for future model development.

Next, we assess the ensemble's discrimination. Figure 15. Extreme Diagnostics of SFNO BVMC and IFS ENS. Diagnostics are averaged over forecasts initialized at 00 UTC for each day in June, July, and August 2023 (total of ninety initialization days). SFNO BVMC is validated against ERA5, and IFS ENS is validated against the ECMWF operational analysis. (a) measures the Receiver Operating Characteristic of the Area Under the Curve. Higher is better. (b) measures the threshold-weighted CRPS. Lower is better. (c) measures the reliability diagram, which compares the forecast probability to the observed occurrence. Reliable ensemble forecasts appear along the one to one line.

is a continuous value indicating the proportion of the ensemble that predicts extreme. This performance is aggregated over all grid cells and all initial times of summer 2023. This results in a reliability diagram, calculated for lead times of 120 hours and 240 hours. We visualize the reliability diagrams at other lead times in the Supplemental Figure C1.

Figure 15b shows that SFNO-BVMC and IFS ENS have a comparable ability to discriminate between extremes and nonextremes\_non-extremes. Both ensembles have similar ROC Area Under Curve (AUC) scores, which measure the discrimination of an ensemble. The ROC curve varies the threshold for classifying an event as "extreme" or "not extreme" from 0 to 1: for each threshold, the resulting true positive and false positive rates are plotted. A successful ROC curve would have a 0 false positive rate and 1 true positive rate: the area under such a curve would be 1. To calculate the ROC AUC scores in Figure 15b, we use the EFI. To actually compare the EFI to observations, EFI ROC curves serve as ECMWF's Supplemental Score on Extremes in their IFS validation (Haiden et al., 2023). The IFS EFI is defined on a daily mean temperature, not a six-hourly temperature. Therefore, in the EFI ROC AUC score in Figure 15b uses a threshold based on daily means. This results in 12 thresholds for extreme weather (one for each month), instead of 48 thresholds (one for each month for each time of day, as in (Price et al., 2023)). Based on the available data on the ECMWF MARS data server, we can only access IFS EFI values until a lead time of

\_7 days, so we only show IFS scores up to that lead time. At long lead times (approaching 14 days), much of the SFNO-BVMC EFI skill comes from the strong El Niño in summer 2023. Because the EFI is only calculated on data with a daily sampling

frequency, Figure 15b necessitated a different extreme threshold than Figures 15a and c. This difference is necessary to enable comparison of EFI ROC curves with Haiden et al. (2023) and extreme diagnostics with Price et al. (2023).

#### 3.3.3 Threshold-weighted Continuous Ranked Probability Score

We calculate threshold-weighted CRPS (twCRPS) on SFNO-BVMC and IFS ENS as a summary score. ExtremeSince extreme weather events have tremendous societal consequences. Therefore, a natural goal is to validate these weather forecasts specifically on their performance for such extremes. One approach might be to evaluate the forecasts during times of extreme weather. However, Lerch et al. (2017) explain the concept of the forecaster's dilemma, which is a common pitfall that occurs with this strategy. This dilemma occurs when a forecast is validated on its extreme event forecasts only when those extremes actually happen. With this verification setup, a forecast system can hedge its performance by overpredicting extreme events. Since it iswould never be evaluated during common weather, the forecast would not be penalized for its overly extreme predictions. By construction, statistically proper scoring rules do not allow for such hedging, and twCRPS is one such scoring rule (Gneiting and Ranjan, 2011; Allen et al., 2023).

The equation for twCRPS is

$$(F,y,w) = \int\limits_{-\infty}^{\infty} (F(z) - 1\{y \le z\})^2 w(z) dz \qquad (F,y,w) = \int\limits_{-\infty}^{\infty} (F(z) - 1\{y \le z\})^2 w(z) dz \qquad \text{twCRPS}$$
 
$$= E_F |v(X) - v(y)| - \frac{1}{2} E_F |v(X) - v(X')| \qquad = E_F |v(X) - v(y)| - \frac{1}{2} E_F |v(X) - v(X')| \qquad \text{(3)}$$

where w is a weighing function, X is a random variable drawn from the ensemble distribution, y is the verification value, and v is the antiderivative of w. We refer the reader to Allen et al. (2022) for further discussion of twCRPS- and its derivation. We choose a weighing function

20 otherwise

This weighing function is applied at each grid cell. t is the 95th percentile 2m temperature described above; it is, calculated for each time of day for each month.

Equation 3 and Equation 1 have the same structure; the difference is that Equation 3 applies v to X and y. Therefore, twCRPS reduces to calculating the standard CRPS score, when the ensemble and the ground truth are transformed using the following function (Allen et al., 2022):

$$twCRPS(F_y) = CRPS(F_t max(y,t))$$
(5)

where  $F_t$  is the CDF of the transformed ensemble. The transformed ensemble isfor each ensemble member  $E_i$ , where i goes from 1 to N for an ensemble size of N. As, the transformed ensemble member is

$$E_{i}^{'} = \max(E_{i}, t) \tag{6}$$

The CDF of the transformed ensemble,  $F_t(x)$ , is thus calculated as  $\frac{1}{N} \sum_{i=1}^{N} 1(\max(E_i, t) \le x)$ . This transformation is described in further detail in  $\frac{1}{N} \sum_{i=1}^{N} 1(\max(E_i, t) \le x)$ , (Allen et al.-(., 2022).

Similar to CRPS, twCRPS is calculated independently for each grid cell, for each forecast initial time. After taking a global average and an average over each initial time in summer 2023, the twCRPS scores are shown as a function of lead time in Figure 15b.

twCRPS assigns no penalty when the ensemble forecast and the ground truth are below the extreme threshold. This is the most common situation that accounts for much of the CRPS score, but it can mask out the performance on extremes. If an ensemble member lies above the threshold when the truth is below the threshold, then the ensemble will be penalized with a higher twCRPS. This is a solution to the forecaster's dilemma: the ensemble can no longer hedge its score by overpredicting extreme events above the threshold. If an ensemble forecast is below the threshold while the truth is above the threshold (false negative extreme), then the ensemble is also penalized. As the ensemble is transformed according to Equation 56, this penalty is determined by the distance between the threshold and the truth, not the distance between the raw forecast and the truth. Therefore, twCRPS penalizes both overprediction and underprediction of extremes. It provides the benefits of the standard CRPS score, as it evaluates a probabilistic forecast of a single ground truth value.

Figure 15b shows that SFNO-BVMC and IFS have similar twCRPS scores. In fact,, with SFNO-BVMC outperforms IFS, with a lower twCRPS scoreperforming slightly better on this metric. Since this score assesses the prediction of the tails of the distribution, SFNO-BVMC is a trustworthy model for predicting extreme 2m temperature events. The twCRPS has the same units as the standard CRPS; for 2m temperature, the units are degrees Kelvin. However, the values for twCRPS are lower than those for CPRS because the former assigns no penalty for the most common case, when both the ensemble members and the ground truth value are below the threshold. In those cases, the twCRPS score will be 0. Relatedly, the score will be very close to 0 if most (though not all) of Because the ensemble members predict a non-extreme event and the ground truth is a non-extreme event. This behavior brings down the value of verification are transformed as in Equations 5 and 6, the twCRPS score, compared to has a lower value than the CRPS score.

twCRPS complements other forecast diagnostics, including those specifically focused on extremes. Recently, Ben Bouallègue et al. (2024) validate PanGu weather on extreme weather events, in part by comparing quantile-quantile plots of PanGu, IFS, and ERA5. While these plots compare the aggregate distributions of the forecasts and the truth, they do not assess whether extreme forecasts are collocated (in space and time) with extreme observations. Ben Bouallègue et al. (2024) state that additional diagnostic tools are necessary to evaluate this. We suggest that twCRPS fills this need, as it focuses on

the tails of the ensemble distribution, but it also evaluates whether the forecasts coherently predict extremes at the right space and time.

#### 4 Discussion and Conclusion

In Part I of this two-part paper, we introduce SFNO-BVMC, an entirely ML-based ensemble weather forecasting system. This ensemble is orders of magnitude cheaper than physics models, such as IFS. It enables the creation of massive ensembles that can characterize the statistics of low-likelihood, high-impact extremes. Here, we present the ensemble design, which uses bred vectors as initial condition perturbations and multiple checkpoints as model perturbations. Multiple checkpoints are created by retraining SFNO from scratch, with a different set of random weights when SFNO is first initialized. In this manuscript, we present a range of ensemble design choices and rationale for making these decisions; we list these in Table 1. To maximize dispersion, we use a large SFNO with a small scale factor and large embedding dimension, and we avoid multistep fine-tuning.

\_We assess the fidelity of SFNO-BVMC on overall ensemble diagnostics, spectral diagnostics, and extremes diagnostics. This comprehensive pipeline is specifically designed for ensemble forecasts (not solely for deterministic ones). As the field of ML-based ensemble forecasting rapidly grows, we hope that other groups also use these statistics to evaluate their ensembles. On overall diagnostics, SFNO-BVMC's performance is 18 hours behind IFS ENS, a leading operational weather forecast on most diagnostics and most variables. We present a pipeline to evaluate the ensemble's performance on extreme 2m temperature events.

On overall diagnostics, SFNO-BVMC's performance lags approximately 12–18 hours behind IFS ENS. We present a pipeline to evaluate the ensemble's performance on extreme 2m temperature, 10m wind speed, and heat index events.

The spectral diagnostics demonstrate that individual ensemble members in the SFNO-BVMC have blurry predictions compared to ERA5. We minimize this as much as possible through a small scale factor, a large embedding dimension, and no autoregressive fine-tuning. Still, some degree of blurring still remains. However, our spectral diagnostics reveal that the spectra from SFNO-BVMC remain constant throughout the rollout. This means that SFNO-BVMC's ability to predict extreme weather and fine-scale phenomena remains constant. Additionally, the SFNO-BVMC ensemble-mean spectra indicate that the ensemble members realistically diverge. Future research and architectural improvements are necessary to reduce the extent of the initial blurring.

Bred vectors are open-sourced through the earth2mip package, and they can readily be applied to other deterministic architectures. This enables out-of-the-box ensemble forecasts from the wide array of existing deterministic architectures. Indeed, recently, there have been over twenty deterministic ML weather prediction models (Arcomano et al., 2020; Bi et al., 2023; Nguyen et al., 2023; Chen et al., 2023b; Bodnar et al., 2024; Mitra and Ramavajjala, 2023; Ramavajjala, 2024; Pathak et al.,

2022; Bonev et al., 2023; Weyn et al., 2021; Willard et al., 2024; Keisler, 2022; Karlbauer et al., 2023; Rasp et al., 2024, 2020; Lang et al., 2024; Couairon et al., 2024; Scher and Messori, 2021; Chen et al., 2023a). It is computationally expensive and

programmer time-intensive to convert all these architectures into ensembles using probabilistic training (e.g. through diffusion models or through CRPS-training on the CRPS loss function). Even for the architectures that are converted to probabilistic training, bred vectors and multiple checkpoints can provide baseline ensemble scores. This baseline can be used to guide further development of end-to-end training.

Understanding how ML models respond to perturbations is an important research frontier (Bülte et al., 2024; Selz and Craig,

2023). SomeUnderstanding how ML models respond to perturbations is an important research frontier (Bülte et al., 2024; Selz and Craig, 2023). In particular, future work is necessary to compare the computational cost and skill of different initial condition perturbations perturbation methods (Bülte et al., 2024), in tandem with model perturbations. We find that bred vectors are a computationally inexpensive way to achieve reasonable spread-error ratios and to generate an arbitrarily large ensemble. Further refinement of initial condition perturbation techniques is needed to improve forecast performance. Two advantages of bred vectors are that they do not rely on external sources. For instance, and they can be used to generate arbitrarily large ensembles. First, Price et al. (2023) use external perturbations from operational data assimilation that provide valuable information aboutto include estimates of observational uncertainty. With the PanGu ML model, Bülte et al. (2024) createtest ML ensembles usingwith IFS perturbations, but they find that these perturbations do not lead to the best performance. Other initial condition perturbations, such as bred. Bred vectors, do not rely on external sources. If an ML model is used to emulate climate models (e.g. in Watt-Meyer et al. (2023)), thesebred vector perturbations are still available, unlike IFS or data assimilation perturbations. AdditionallySecond, there are often a limited number of external perturbations from existing weather center. However, bredcenters. Bred vectors can be used to generate arbitrarily large ensembles, such as the huge ensemble in Part II. For operational weather forecasting, future work is necessary to improve calibration by combining multiple types of perturbations.

Looking to the future of ML-based ensemble forecasting, an important design choice is whether the ensemble is created during training or after training. NeuralGCM (Kochkov et al., 2023) and GenCast (Price et al., 2023) create ensembles endtoend during training; they train using probabilistic loss functions. Here, we train SFNO using a deterministic loss function, and we create the ensemble after training. In the machine learning literature, it is an openactive area of research whether ensemble training or post hoc ensembling leads to the most reliable results (Jeffares et al., 2023). In weather forecasting, so far, GenCast and NeuralGCM offer superior ensemble performance to SFNO-BVMC. They have better CRPS scores and spread-skill ratios. Even at full ERA5 horizontal resolution, GenCast does not produce blurry forecasts. While GenCast and SFNO-BVMC run on different hardware (TPUs, compared to NVIDIA GPUs used here), GenCast takes 6 minutes to create a 2-week forecast, with a timestep of 12 hours. At the same horizontal resolution, SFNO-BVMC takes 1 minute to create a 2-week forecast, with a timestep of 6 hours; therefore, SFNO-BVMC appears to be a factor of 12 faster for inference. In part, this difference is because SFNO-BVMC does not require the iterative denoising used by GenCast at each timestep. In Part II of this

paper, we assess the performance of huge ensembles of SFNO-BVMC. A promising area of future research is to explore the behavior of huge ensembles from these other ML-based models.

The current generation of ML-based ensemble weather forecasts all have core design differences. IFS ENS uses physicsbased modeling, NeuralGCM uses a differentiable dynamical core and an ML physics parameterization, GenCast uses a diffusion-based generative model, and SFNO-BVMC uses deterministic training. Because of these differences, future researchwork is necessary to assess the strengths and weaknesses of each model in different meteorological regimes. When different forecasting systems have uncorrelated errors, a multimodel ensemble can lead to improved skill. Each forecasting system could be post-processed, bias-corrected, and optimized to create the best ensembles for each region.

As the use of machine learning and huge ensembles grows in weather forecasting, it is important to consider climate equity (McGovern et al., 2024). Weather forecasts bring tremendous societal and economic value, and it is important to make them as accurate as possible across the global (Linsenmeier and Shrader, 2023). Considerations of forecast skill should be improved for all locations, not just locations with large weather centers. One benefit of SFNO-BVMC is that it creates forecasts at a fraction of the computational cost. This means that organizations with limited access to large supercomputing resources can run weather forecasts and optimize them for their specific end use cases and datasets. In particular, they can be fine-tuned for regional purposes. In this introductory work, we primarily consider global metrics, and we focus on 2m temperature. In the tropics, temperature variance is small due to a smaller Coriolis parameter, and humidity variations are particularly important, especially for impactful rainfall. Future work is necessary to consider the ensemble calibration and performance at the regional level, and this work can include explicit considerations of other variables, such as rainfall and humidity. In particular, the SFNOs trained here do not predict precipitation, and accurate medium-range rainfall forecasts are an important frontier in ML weather research.

In this manuscript, we run our extreme diagnostics pipeline on warm temperature extremes, and we validate on summer 2023, as it is the hottest summer in the observed record. However, future work is necessary to characterize At lead times of 48 hours and 96 hours, the performance on cold temperature extremes and wind extremes is similar to IFS. However, future work is necessary to reduce false positives for these other extreme events of interest (e.g. extreme wind, vapor transport, or precipitation classes of extremes at 10-day lead times (Figure D3). The EFI here is calculated on daily mean temperature, but it can also be calculated for other quantities, such as daily max or min temperature, convective available potential energy, or vapor transport (Lavers et al., 2016). Similarly, the ROC curves and reliability diagrams could be calculated for other types of extremes. We have presented an ensemble extreme diagnostics pipeline that can be used to guide development for other ML data-driven weather systems.

| <del>u10m (m/s)</del>  | 0.52             |
|------------------------|------------------|
| <del>v10m (m/s)</del>  | 0.53             |
| <del>u100m (m/s)</del> | 0.65             |
| <del>v100m (m/s)</del> | 0.66             |
| t2m (K)                | 0.37             |
| sp (Pa)                | 40.80            |
| msl (Pa)               | <del>42.30</del> |

Figure A1. Bred Vector Perturbation Amplitudes. The root mean square amplitude of the perturbation is shown for each variable.

In Part II, we use SFNO-BVMC to generate a huge ensemble, with 7,424 members. This ensemble is 150x larger than the ensembles used for operational weather forecasting. We explore how an ensemble of this size enables analysis of the statistics of low-likelihood low-likelihood, high-impact extremes.

#### A Case Study: 2023 Phoenix Heatwave

We include a case study for a heatwave in Phoenix in summer 2023. Phoenix had temperatures above 310 K (36.85 C) for over 30 consecutive says in summer 2023. We compare the ensemble forecasts from SFNO-BVMC and the IFS ensemble in Figure A1. We compare SFNO-BVMC to the IFS ensemble, and we visualize their respective verification datasets. We show that at a lead time of 3 days, SFNO-BVMC can forecast the high temperatures observed over the region during the region. The IFS ensemble is initialized with an operational analysis, not ERA5, and we use this analysis as the verification dataset for IFS. Notably, the operational analysis has even sharper fields than ERA5: this has previously been quantified in Figure S38 and Supplementary Materials Section 7.5.3 of Lam et al. (2023). The difference between operational analysis and ERA5 reanalysis is shown for this heatwave in Figure A1.

#### **AB** Perturbation Amplitudes

The root-mean-square amplitude of the bred vector perturbations is set to be 0.35 \* the deterministic RMSE of SFNO at 48 hours. Figure A1B1 visualizes the actual numerical value of these amplitudes (with the factor of 0.35 applied).

#### **BC** Definition of Spread and Error

Below, we include our definitions for calculating the spread and error for calculation of the spread-error ratio (Fortin et al., 2014).

The ensemble forecasts have a 0.25 degree horizontal resolution on a regular latitude-longitude grid, so the ensemble forecasts have a 721 latitude points and 1440 longitude points. Let i and j be the indices of a grid cell at a given latitude and longitude.

For each grid cell, the ensemble mean is

$$\mu(i,j) = \frac{1}{N} \sum_{n=1}^{N} x_n(i,j)$$

For each grid cell, the ensemble variance is

$$\sigma^{2}(i,j) = \frac{1}{N} \sum_{n=1}^{N} (x_{n}(i,j) - \mu(i,j))^{2}$$

To calculate the spread in the spread-error ratio, we first calculate the ensemble variance at each grid cell. Then, we take the global latitude-weighted mean of this variance. Then, we take the mean over forecasts from multiple initial dates. Finally, we take the square root.

Figure A1. 2023 Phoenix Heatwave. Comparison of ensemble forecasts from SFNO-BVMC (a) and the IFS ensemble (b) at a grid cell near Phoenix, Arizona, USA. Both models' forecasts are initialized on June 27, 2023 at 00:00 UTC. The SFNO-BVMC verification dataset is ERA5 and the IFS ENS verification dataset is operational analysis. (c) and (e) show ERA5 and operational analysis, respectively, for the daily max temperature on June 30, 2023. (d) and (f) show the SFNO-BVMC ensemble mean and IFS ENS mean, respectively, for the daily max temperature on June 30, 2023. The black stars in c-f denote the grid cell near Phoenix, Arizona, USA used in (a) and (b).

| <u>u10m (m/s)</u> | 0.52  |
|-------------------|-------|
| v10m (m/s)        | 0.53  |
| u100m (m/s)       | 0.65  |
| v100m (m/s)       | 0.66  |
| t2m (K)           | 0.37  |
| sp (Pa)           | 40.80 |
| msl (Pa)          | 42.39 |
| tcwv (mm)         | 0.95  |
| d2m (K)           | 0.47  |
|                   |       |

Variable Amplitude

Figure B1. Bred Vector Perturbation Amplitudes. The root-mean-square amplitude of the perturbation is shown for each variable.

where l(i,j) denotes the latitude weight for grid cell i,j. The latitude weights enable calculation of the global mean.

We follow a similar process for calculating the error, except the ensemble variance is replaced with ensemble mean-squared error.

$$= \sqrt{\frac{1}{\mathcal{T}} \sum_{t=1}^{\mathcal{T}} \sum_{i=1}^{721} \sum_{j=1}^{1440} l(i,j) (\mu(i,j) - y)^2}$$
 Error

where y denotes the verification value. The spread and error are calculated for each lead time and shown in Figure 8.

Results

are shown for all forecasts in the test set year, 2020, so T = 732, for 732 initial times (2 per day).

## D Diagnostics for Additional Variables and Lead Times

We show the reliability of the forecasts from SFNO-BVMC at a lead time of two days and four days in Figure D1. IFS is more reliable than SFNO, since its forecasts lie closer to the 1-to-1 line, though the performance is comparable. When SFNO-BVMC predicts 95th percentile temperature with approximately 20 to 30% probability, the actual occurrence is more frequent than this predicted probability.

We also include the reliability diagrams for the heat index (Lu and Romps, 2022), which combines 2m temperature and moisture, and for 10m wind speed. In Figure D2, we show the reliability diagrams for 95th percentile heat index. As for 2m air temperature, the 95th percentile heat index is calculated from 1993-2016 ERA5 climatology. Comparing Figure D2 to Figure 15, we find that SFNO-BVMC is similarly reliable for heat index extremes as it is for warm 2m temperature extremes.

Next, we show overall CRPS scores and reliability diagrams for 10m wind speed, and we also show the model's reliability for forecasting cold extremes (5th percentile 2m temperature). To calculate these statistics, we use December-January-

February (DJF) from 2021 to 2022, while we use summer 2023 in Figure 15. We use a winter season for the Northern Hemisphere to include sufficiently cold extremes over land in the Northern Hemisphere. Additionally, DJF 21-22 is within the time period included by the WeatherBench dataset (Rasp et al., 2024), so we can readily access the IFS ensemble wind forecasts via a Zarr file stored on Google Cloud, without having to download additional data from ECMWF's tape servers. On 10m wind speed and cold extreme air temperature, SFNO-BVMC and the IFS ensemble have similar performance on overall CRPS

#### Reliability at Predicting 95th Percentile 2m Temperature

Figure <u>C4D1</u>. Reliability Diagram at 48-hour and 96-hour lead times. Reliability diagrams are shown for 95th percentile extremes at a lead time of 48 hours and 98 hours. Reliability diagrams are calculated using all initial times from summer 2023. Successful forecasts lie along the 1-to-1 line.

Figure D2. Reliability Diagram for Heat Index. The heat index combines 2m air temperature and 2m dewpoint. Reliability diagrams are shown for 95th percentile heat index extremes at a lead time of 48 hours and 98 hours (a) and 120 and 240 hours (b). Reliability diagrams are calculated using summer 2023 forecasts, from June 1, 2023 to Aug 14, 2023.

and on reliability diagrams (Figure D3). This indicates that the ensemble generation methodology (bred vectors and multiple checkpoints) is promising for other variables and classes of extremes. SFNO-BVMC does degrade in reliability when making

Figure D3. SFNO-BVMC Performance on 10m Wind Speed and Cold Extremes. (a) Overall CRPS for SFNO-BVMC and the IFS ensemble on 10m wind speed. Lower CRPS scores are better (b) Reliability Diagrams for 95th Percentile 10m wind speed for SFNO-BVMC and IFS at lead times of 48 and 96 hours. (c) Reliability Diagrams for 5th percentile temperature extremes for SFNO-BVMC and IFS at 48 and 96 hour lead times of 48 and 96 hours. All scores are calculated using all forecasts initialized in December-January-February 2021. Successful forecasts lie along the 1-to-1 line.

forecasts of extreme wind with 90% probability. This problem is accentuated at longer lead times (see below), and future research is necessary to isolate the cause of this behavior.

We identify two areas for future research and model improvement. First, interestingly, both SFNO-BVMC and the IFS ensemble have degraded performance for forecasting cold extremes, as opposed to warm extremes (compare the 5th percentile reliability in Figure D3 to the 95th percentile reliability diagrams in Figure D1). With future model development, we hope to improve the performance of SFNO-BVMC in forecasting cold extremes. Second, for 10m wind speed and cold temperature extremes at a lead time of 10 days, SFNO-BVMC's reliability degrades (Figure D4). For these variables, the ensemble still has a good overall forecast scores (see the wind speed CRPS in Figure D3 and the 2m temperature Figure 7). SFNO-BVMC reliability is close to IFS for extreme forecast probabilities from 0% to 50%. However, the reliability drops when the model predicts high probabilities (greater than 70%) of extreme conditions. In these cases, SFNO-BVMC tends to be overconfident: its forecast of an extreme event does not match the observed outcome. This overconfidence occurs extremely rarely. At a lead time of 10 days, it is very uncommon (less than 1% of all forecasts for 2m temperature, less than 0.1% for 10m wind speed) for SFNO-BVMC to predict a greater than 70% chance of extreme wind or cold temperatures. At this long lead time, there is significant ensemble spread induced by the perturbations, so the ensemble system is not confident in issuing extreme forecasts. Still, having a calibrated reliability diagram is crucial for all forecast probabilities, and this shortcoming must be resolved with future model development.

#### E Error in Computer Code For Bred Vector Calculation

After performing the analysis in this manuscript, we discovered an error in our calculation of bred vectors. During the first steps of calculating bred vectors at  $t_{-2}$  and  $t_{-1}$  (Figure 4), we incorrectly supplied SFNO with the solar zenith angle at time  $t_0$ .

We

Figure D4. Reliability Diagram at 240h Lead Time for 95th percentile wind speed and 5th percentile temperature extremes. Reliability diagrams are shown for 95th percentile extremes at a lead time of 48 hours and 98 hours. Reliability diagrams are calculated using all forecasts initialized in December-January-February 2021. Successful forecasts lie along the 1-to-1 line.

Figure D5. Perturbed Spectra at 0h. Same as Figure 11, but showing the spectra of perturbations applied to the ERA5 initial conditions.

have verified and established that this error does not affect any of the conclusions or scores presented in this manuscript. This error does not make a discernible difference for three reasons. First, the last breeding step calculates the perturbations using SFNO at  $t_0$ . At  $t_0$ , the correct zenith angle is supplied, so the final perturbation is still based on the correct SFNO forecasts. Second, we validate that the error does not cause undesired artifacts related to the diurnal cycle in the actual perturbations (see Figure 5). Third, the breeding cycle only uses 1-step forecasts, which means that the error from using the incorrect zenith angle does not grow.

Figure E1. CRPS Comparison of Original and Fixed Bred Vector Perturbation Method. Our original calculation of bred vectors contained an error with a mismatched cosine zenith angle during the first two breeding steps. This figure compares the CRPS of an ensemble with the "Original" (incorrect) bred vector calculation and the "Fixed" calculation for 2m and 850hPa atmospheric temperatures, the 500hPa geopotential height, and the 850hPa zonal wind as representative fields. Results are shown for 52 forecasts initialized in summer 2020, one per week starting January 2.

We also note that this error does not affect the 15-day rollout of SFNO, only the calculation of the bred vectors. Given the nature of this error, we do not believe it would cause SFNO-BVMC to appear better than it actually is. Instead, it would be more likely to degrade its performance, making the method seem worse than it really is. We compare the ensemble scores for the ensemble with the error (named "Original") and the fixed ensemble (named "Fixed") in Figures E1 and E2. We have corrected the error in the GitHub page for our project, but for scientific reproducibility, the error remains in the codebase in the DOI in our Code and Data Availability Section, since this is the version of the code that we used for our analysis.

Author contributions. Bold words correspond to Contributor Roles Taxonomy (CrediT) conventions. AM and WDC contributed equally to this work. AM, BB, NB, JE, YC, PH, TK, JN, TAO, MR, DP, SS, and JW wrote Software and performed Formal Data Analysis. WDC, KK, and MP supervised the research project. WDC, KK, and MP Acquired Funding for the project. WDC, KK, PH, SS, AM, and MP obtained computational Resources for the project. All authors contributed to the Methodology of the project. WDC, AM, BB, YC, PH, KK, JN, TAO, MP, MR, SS, and JW contributed to the Conceptualization of the project.

Competing interests. At least one of the (co-)authors is a member of the editorial board of Geoscientific Model Development.

Figure E2. Ensemble Mean RMSE and Ensemble Spread Comparison of Original and Fixed Bred Vector Perturbation Method. Our original calculation of bred vectors contained an error with a mismatched cosine zenith angle during the first two breeding steps. This figure compares the ensemble mean RMSE and ensemble spread of an ensemble with the "Original" (incorrect) bred vector calculation, and the "Fixed" calculation. Results are shown for 52 forecasts initialized in summer 2020, one per week starting January 2.

Acknowledgements. This research was supported by the Director, Office of Science, Office of Biological and Environmental Research of the U.S. Department of Energy under Contract No. DE-AC02-05CH11231 and by the Regional and Global Model Analysis Program area within the Earth and Environmental Systems Modeling Program. The research used resources of the National Energy Research Scientific Computing Center (NERSC), also supported by the Office of Science of the U.S. Department of Energy, under Contract No. DE-AC02-05CH11231.

The computation for this paper was supported in part by the DOE Advanced Scientific Computing Research (ASCR) Leadership Computing Challenge (ALCC) 2023-2024 award 'Huge Ensembles of Weather Extremes using the Fourier Forecasting Neural Network' to William Collins (LBNL). This research was also supported in part by the Environmental Resilience Institute, funded by Indiana University's Prepared for Environmental Change Grand Challenge initiative.

Code and data availability. The code, datasets, and models are all stored at https://doi.org/10.5061/dryad.2rbnzs80n-, via DataDryad. The code is integrated with Zenodo: https://doi.org/10.5281/zenodo.14710345 at the prior DOI. We include the code to train SFNO, conduct ensemble inference with bred vectors and multiple checkpoints, and scoring and analysis code. We also open-source the model weights of the trained SFNO. See the README of the former DOI for information on how to use the codebase and for the permissive license associated with the code and data. The code is available via the Lawrence Berkeley Lab BSD variant license, and the data is available with a CCO license. For convenience, the webpage of our project is https://github.com/ankurmahesh/earth2mip-fork.

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
