# Peer review of "Huge Ensembles Part I: Design of Ensemble Weather Forecasts using Spherical Fourier Neural Operators"

_EGUsphere, 2024_

## Author Response (AR1)

**Response to Reviewers: "Huge Ensembles Part I: Design of Ensemble Weather Forecasts using Spherical Fourier Neural Operators"**

February 17, 2025

**Overview**

We sincerely thank the reviewers for their constructive comments and review of our paper. These comments will substantively improve our manuscript. We have included responses to the reviewers' comments below, with the reviewer comments in black text and our response in green text.

In this document, we will detail our planned revisions. Some of the revisions require re-analyzing our ensemble simulation with additional variables, such as 10m north-south wind, 10m east-west wind, and heat indices. The ensemble simulations use O(1) TB per variable for a 58-member ensemble, and O(100) TB per variable for the huge ensemble. Upon completion of the analysis that the reviewers suggest, we will submit a revised version of the manuscript in mid-March 2025.

For public reference, this is the first of a two-part manuscript on huge ensembles. We refer to part I as HENS Part I [Mahesh et al., 2024a], and we refer to part II has HENS Part II [Mahesh et al., 2024b].

**Comments from Reviewer #1**

Part 1 and 2 are both interesting papers that document the development and use of a machine learned ensemble weather forecast model with an enormous number of ensemble members. We request a short period of time due to the computational and data requirements of these comments. The papers fit well into GMD, but I think that they should be revised following the comments below. The paper is documenting very interesting results, as it shows that an SFNO-type-model can be used to develop a competitive ensemble forecast system when combined with bred vectors and multi-check pointing.

Thank you very much for your review of our paper.

Page 2: The ML model has "orders-of-magnitudes" lower computational cost. Is this really true? More than a factor of 10? This could only be possible if the IO cost (that will stay the same) is considered to be of less than 10% of the overall cost (also see comment for Part 2). And what is the "cost"? Time, energy, or hardware purchase?

Thank you for raising this concern. We are not considering I/O costs here and are only basing this off an estimate of time to generate an ensemble member. Due to differences in hardware (CPU vs. GPU), high-performance computing centers, network, energy requirements, data storage availability/ costs, hardware costs, time step, and spatial resolution, we intentionally do not provide a comprehensive breakdown and comparison between SFNO and IFS in this work. The basis of our statement is that with **96** CPUs and **60**

minutes, we can generate **1** IFS ensemble member. However, with **1** GPU and **60** minutes, we can generate **60** ensemble members with SFNO. Conservatively, this represents a 60x decrease in the time required to generate the ensemble member. It is a significantly larger decrease if we consider the CPU time for IFS vs. the GPU time for SFNO. We of course acknowledge that CPUs and GPUs are different computing platforms. Physics-based models could also be sped up on GPUs, though such versions are not yet available in many cases. However, our goal is not to benchmark the compute complexity and required floating point operations for physics-based models and ML models in this work. Instead, we demonstrate that currently, it is more feasible and practical to run huge ensembles with SFNO, since SFNO takes much less time to generate 1 member per GPU than IFS does per CPU. We simply wish to illustrate that running SFNO is computationally cheap on GPUs, similar to the statements made here regarding ECMWF's data-driven weather model [Alexe et al., 2024]. We discuss this more in our Part 2 peer review response.

For our group, another practical consideration is that our compute time is primarily measured on a per-node basis, not a per-GPU basis. Since we have 4 GPUs per node on Perlmutter, we can increase the throughput of the number of ensemble members by a factor of 4. (Of course, we fully acknowledge that this situation varies based on the inference setup of different users, and this varies widely based on users. However, it was a relevant practical consideration for our group when we decided whether to run huge ensembles with SFNO or with a physics-based model.)

P6, paragraph starting with "We choose SFNO...": I find this part difficult to follow. It would be good to remind the reader about the architecture of the SFNO and to clearly state what is changed to see only "linear" scaling with horizontal resolution. This will clearly not be the case if the size of the SFNO is increased (?). And I thought I had seen talks by NVIDIAns that showed that there were fundamental problems when scaling SFNO to km-scale resolution? And is "super-linear" more or less than linear when you talk about the cost?

Thank you for pointing out these issues. We have now removed the section on how SFNO scales with input resolution. We believe that this discussion on compute complexity would best happen in research that performs global, kilometer-scale emulation, particularly after any necessary changes to model size and hyperparameters (e.g. embedding dimension and scale factor) have been made. We are not performing this kilometer-scale emulation in HENS Part I and Part II, so we believe it would be best to save this topic for future work.

What exactly do you mean by downscaling and scale factor here (I think I know, but only since I know the previous papers)? I do not understand why a lower scale factor would lead to a larger ensemble spread.

In SFNO, the input is encoded into a latent representation. In the latent space, the SFNO blocks process the latent representation to optimally predict the next time step. Then, this transformed latent input is decoded to the physical fields of the next input prediction. The scale factor is the parameter that determines the resolution of the latent representation of the input. A scale factor of 1 would mean that the latent representation has the same dimensions as the input. A scale factor of 2 downsamples the input by a factor of 2, and so forth. We have modified the text to include this.

Figures 4e-h in Brenowitz et al. [2024] show that models with lower scale factors have better spread-error ratios. Additionally, we illustrate that the bigger models (lower scale factor and also embed dimension) have better spread-error ratios in Figure 2a. These bigger models also have less spectral degradation (Figure 2b). They are less blurry. We hypothesize that they blur out less of the initial condition perturbations; instead, they better incorporate the finer scales of the perturbations in their predictions. We speculate that this could lead to more ensemble spread, in part by allowing for more transfer of perturbation energy from small scales

77 to larger scales.

78    Figure 4: I do not really understand how the process is repeated for t-1 and t0.

79 Thank you for making us aware of this, and we apologize for any confusion. We use the following process:
80 at $t_{-3}$, add random spherical noise to Z500, and compute the difference between the perturbed forecast and
81 a control forecast. We use this difference as the new perturbation, and we add it to the state at $t_{-2}$. Again,
82 we calculate the difference between the perturbed state and the control state, which would result in a new
83 perturbation. For $t_{-1}$, we continue with the same algorithm. We add the newly calculated perturbation to
84 $t_{-1}$, and we calculate the difference between a perturbed forecast and control forecast. Here is an algorithm
85 to help clarify. For input state $X$ (the 74-channels compromising the state of SFNO), spherical random noise
86 $\epsilon$, and bred vector perturbation $\delta$,
* * *
**Algorithm 1** Bred Vector Algorithm
* * *
1: Add random spherical noise $\epsilon$ to $\mathbf{X}_{t_{-3}}$ Z500:

$$\mathbf{X}_{t_{-3}}^{\text{perturbed}} = \mathbf{X}_{t_{-3}} + \epsilon$$

2: Compute difference between perturbed and control forecasts at $t_{-3}$:

$$\delta = \text{SFNO}(\mathbf{X}_{t_{-3}}^{\text{perturbed}}) - \text{SFNO}(\mathbf{X}_{t_{-3}})$$

3: Add $\delta$ as the new perturbation. Add to the state at $t_{-2}$:

$$\mathbf{X}_{t_{-2}}^{\text{perturbed}} = \mathbf{X}_{t_{-2}} + \delta$$

4: Compute difference between perturbed and control states at $t_{-2}$:

$$\delta = \text{SFNO}(\mathbf{X}_{t_{-2}}^{\text{perturbed}}) - \text{SFNO}(\mathbf{X}_{t_{-2}})$$

5: Repeat the process for $t_{-1}$:

$$\mathbf{X}_{t_{-1}}^{\text{perturbed}} = \mathbf{X}_{t_{-1}} + \delta$$

$$\delta = \text{SFNO}(\mathbf{X}_{t_{-1}}^{\text{perturbed}}) - \text{SFNO}(\mathbf{X}_{t_{-1}})$$

6: Repeat the process for $t_0$:

$$\mathbf{X}_{t_0}^{\text{perturbed}} = \mathbf{X}_{t_0} + \delta$$

$$\delta = \text{SFNO}(\mathbf{X}_{t_0}^{\text{perturbed}}) - \text{SFNO}(\mathbf{X}_{t_0})$$

7: **Output:** Bred vector perturbation $\delta$
* * *
87    Figure 10: Is the control member equivalent to a normal ensemble member, or are there small differences
88 (as in IFS)?

[Figure]

Peer Review Figure 1: ECMWF M-Climate Description. See the second bullet point for a description of the number of initial dates used. ECMWF M-Climate was taken from https://confluence.ecmwf.int/display/FUG/Section+5.3.1+M-climate%2C+the+medium+range+model+climate. (Date accessed: 02/17/2025)

We have added the control member spectra to this figure and compared it to the spectra range from the perturbed members.

Can you also plot 0h?

We will add the 0h spectra figure to Figure 10 and will present this in the revised manuscript.

Page 20: 9 initial dates per year does not seem correct.

We have confirmed the M-Climate on ECMWF's confluence webpage indicates that the M-Climate for a given initial date consists of the 9 initial dates per year in the stored hindcasts that are closest to that date. For clarity, we include a screenshot of the documentation in Peer Review Figure 1. We archive this screenshot in this document only because we understand that this documentation may change. A similar estimate of 9 initial dates is given in Section 2.1 of Lavers et al. [2016].

Section 3.3.2 and 3.3.3 read a bit too much like a textbook. Can you refer to literature and keep the discussion shorter?

We have shortened and changed the discussion. For clarity, the revisions will be easiest to see via a change log, to which we will point upon submission of the revised manuscript.

Page 25: The discussion of the pipeline and the earth2mip package indicates that you consider this to be one of the main contributions of the paper. I think it could be, but you would probably need to make it more prominent in the write-up. It is hardly mentioned at the moment. You could maybe show how the package is working for another ML model, more-or-less out of the box?

Thank you for this suggestion. The earth2mip package was made independent of and prior to our manuscript. We heavily use this pre-existing package for inference and scoring, and we introduce modifications to support bred vectors and multiple ensembles. The package already includes support for multiple other models, including Graphcast, DLWP, the Adaptive Fourier Neural Operator, and Pangu. However, since the package

is more general-purpose than our manuscript, and because it was made independent of the manuscript, we do not include these demonstrations here. We note that the package already includes examples of using other models at (see https://nvidia.github.io/earth2mip/examples/index.html, Date Accessed: 02/16/2025).

Is the extreme diagnostics pipeline that is mentioned on page 27 meant to be used by other groups and to work as a benchmark?

The package includes instructions on integrating other model architectures for inference. The extreme diagnostics pipeline is integrated with our fork of earth2mip. The diagnostics can then be run with other architectures. We would encourage others to use our extreme diagnostics or adapt the code for them into their own workflows, and we open-source these diagnostics to aid this goal. However, for this manuscript, we do not aim to provide an operational platform with a real-time leaderboard on extreme diagnostics, as that is out of scope. Furthermore, the WeatherBench 2 platform already exists with a similar purpose and is a more mature leaderboard.

Abstract: "these" to "These"

Done.

P4: "set This"

Fixed.

Figure 2b: For what timestep are the spectra calculated?

They are calculated at 360 hours. We have added this information to the figure caption.

**Comments from Reviewer #2**

The manuscript presents an approach to forecasting low-likelihood high-impact extreme weather events using a Spherical Fourier Neural Operator with bred vectors and multiple checkpoints (SFNO-BVMC), addressing a significant challenge faced by current deep-learning weather prediction models. The results demonstrate the model's capability to predict extreme events while achieving reduced computational costs compared to traditional Numerical Weather Prediction (NWP) methods, potentially marking a significant milestone in weather forecasting.

Thank you very much for your review and for providing these overall comments.

Despite these promising results, several aspects warrant further research. The authors mainly focused on 2m temperature, especially heat extreme from model configuration to diagnostics, and given that the authors deliberately included 2m dewpoint temperature as a model input variable, incorporating predictions of derived heat extreme indices would provide valuable insights into the model's capabilities. Furthermore, I recommend evaluating a broader range of LLHIs to strengthen the reliability of the approach. I think the authors can incorporate cold extremes along with heat extremes. What about wind extremes, which are in prediction variables?

Thank you very much for these suggestions to help improve our manuscript. In our revised manuscript, we will include the following additions: SFNO-BVMC diagnostics for the heat index, wind, total column water vapor, and cold extremes. We will also include IFS diagnostics on the heat index and wind. We will submit a revised manuscript, and we will provide pointers to these updates for clarity.

Most importantly, this model does not encompass floods/precipitation, which can cause the highest impact extreme. In its current form, the LLHI diagnosis may be too narrow to adequately showcase the model's full ability to predict various extreme weather events.

Thank you for raising this issue. Precipitation is excluded as a variable because of the challenges in obtaining a global training dataset with high-spatiotemporal resolution. Some ML model groups have a "lack of confidence in the quality of ERA5 precipitation data" [Price et al., 2024] and exclude the precipitation results from the primary evaluation [Lam et al., 2023]. In addition to the training dataset challenge, the spatial statistics and long tails of precipitation indicate that further architectural changes may be necessary for some architectures [Pathak et al., 2022]. Therefore, precipitation is not one of the variables in the original SFNO [Bonev et al., 2023]. We note that the exclusion of precipitation is a common feature across many data-driven weather prediction models [Bi et al., 2023, Keisler, 2022, Chen et al., 2023a,b, Ramavajjala, 2024, Cachay et al., 2024, Bodnar et al., 2024], many of which are leading models listed on WeatherBench 2. The addition of precipitation is very much an important challenge at the forefront of data-driven weather prediction. In future research, we certainly wish to emulate precipitation to forecast LLHI precipitation events and will include it as a variable in our ensemble: however, for this work, we focus on the development of ensembles and study extreme surface temperature events (with other variables forthcoming).

As well as various extreme events, actual forecasts would be helpful to recognize the usefulness of the model. Diagnostics with real-event prediction would be more persuasive. For example, t2m ensemble time series at a certain grid point, trajectories of each ensemble for each variable, and the difference between IFS could strengthen the model's credibility.

We have currently presented two demos of actual forecasts in HENS Part II of our manuscript. in Figure 5b of HENS Part II, we compare our ensemble to the t2m distribution to IFS for the 2023 heatwave in Kansas City, Missouri, USA. In Figure 5a, we show how the ensemble range varies for the heatwave prediction as a function of lead time. In HENS Part II Figure 10, we also provide a demo of the distribution of predictions for a heatwave in Shreveport, Louisiana, USA. While these demos are currently in HENS Part II, we will supplement them by adding more related info to this manuscript in HENS Part I, including your suggestion of the ensemble trajectories from SFNO-BVMC and IFS. We sincerely thank you for this recommendation.

Major Comments 1. (p.4) "Existing work has shown that simple Gaussian perturbations do not yield a sufficiently dispersive ensemble. (Scher and Messori, 2021; Bülte et al., 2024): the ensemble spread from these perturbations is too small." If so, you can still adopt singular vectors or other methods to reflect initial condition uncertainty. Are bred vectors superior to other approaches? Are they the cheapest way other than simple Gaussian perturbations?

Bülte et al. include benchmark performance using the IFS perturbations, which include a component from singular vectors. It is well out of the scope of this paper to compare other possible methods of perturbing the initial conditions: we show that bred vectors offer satisfactory performance for our purposes.

2. (p.4) "Each resulting checkpoint represents an equivalently plausible set of weights that can model the time evolution of the atmosphere from an initial state." : (Bonev et al., 2023) assessed their SFNO for weather prediction via ACC only. As hyperparameters and input variables changed, I am curious about the predictability of this version. Does each of the checkpoints generate reliable forecasts? Comprehensive assessment of SFNO via metrics more than ACC is required.

Thank you very much for raising this issue, and we agree that a comprehensive assessment is necessary. In this manuscript, we benchmark SFNO with overall diagnostics (root mean squared error, CRPS), spectral diagnostics (perturbed spectra, control spectra, and spectra of the ensemble mean), and extreme diagnostics (reliability diagrams, Receiver Operating Characteristic curves, threshold-weighted CRPS, and [in Part II] outcome-weighted CRPS). In particular, for assessing reliability, spread-error ratios and reliability diagrams assess whether the forecasts are "reliable," defined as whether the "observed frequency of the event, for a given forecast probability, is equal to the forecast probability" [Johnson and Bowler, 2009]. Some of our diagnostics assess a given SFNO checkpoint (e.g. lagged ensembles, perturbed spectra, control spectra), and the others assess the behavior of all 29 SFNO checkpoints as an ensemble. We believe these benchmarks constitute a comprehensive assessment of SFNO as a single checkpoint but more importantly as an ensemble system. Are there other specific metrics or aspects of SFNO that you wish to see tested? We hope to ensure that our results are rigorous. Thank you for your response and consideration.

3. (p.6) "In this study, we add 2-meter (2m) dewpoint temperature as another variable; for our SFNO training dataset, we obtain the 2m dewpoint temperature field from ERA5.": Vertical velocity and precipitation are excluded. As I mentioned above, precipitation is important in extreme weather forecasting. Is there any specific reason for excluding precipitation?

[See response above.]

4. (p.10) Figure 3. The ensemble spread from different numbers of checkpoints. : Model configuration also focused on 2m temperature. Do we need to change the number of checkpoints if we want to forecast wind extremes? Do we need to change it every time for different variables? Selecting the number of checkpoints based on the comparison among multiple variables would be a more optimal choice.

Thanks for this question. We clarify that it is not intended to change the number of checkpoints in the ensemble based on the user's variable of interest. We will change Figure 3 to show how spread changes as a function of ensemble size for other variables of interest.

We answer question 6 and question 5 in reverse order, for clarity and flow.

6. (p.17) "While the control and perturbed spectra remain constant through the rollout, the SFNO-BVMC ensemble mean does increasingly blur with lead time. Figure 12 shows that the ensemble means of SFNO-BVMC and IFS ENS similarly degrade in power after 24 hours, 120 hours, and 240 hours.": In the first paragraph of section 3.2 Spectral Diagnostics, the authors elaborate that power decay is one of the symptoms of blurriness, but this sentence seems like presuming those two are equivalent. section 3.2 needs to be more clear. What is the relationship between spectra and blurriness in general?

Thanks for this question: we apologize for the confusion. We use spectra as a **measure** of blurriness: a degraded spectrum is a sign that the model output is blurry.

and what did SFNO find?

During the rollout, it is preferable if the spectra of each ensemble member stays constant. This means that each ensemble member realistically represents the atmosphere. When we train SFNO with multistep fine-tuning, the spectra of each member does stay constant. A caveat is that SFNO-BVMC's predictions still

have some blurriness, but at least this level of blurriness stays constant during the rollout.

During the rollout, it is preferable if the spectra of the ensemble mean degrades realistically. As each individual member undergoes a different trajectory, the ensemble mean of all members should get blurrier: the ensemble members should increasingly spread with lead time. We find that the spectra of the ensemble mean from SFNO-BVMC degrades similarly to IFS, a realistic benchmark weather model. This passes a test laid out by Bonavita [2024]

Why is SFNO-BVMC different from other DLWPs regarding the power spectrum?

During the rollout of some deterministic DLWPs, such as Graphcast and the deterministic AIFS, the spectra of each individual ensemble member increasingly degrades with lead time [Lang et al., 2024, Kochkov et al., 2024]. This is not desirable. These other DLWPs not only start off slightly blurry (just like SFNO-BVMC), but they get increasingly blurrier during the rollout.

5. (p.17) "On the second criterion, crucially, their spectra remain constant through the 360-hour rollout (Figure 10 and Figure 11).": Degradation of power in short wavelengths occurs in a lot of DLWPs. Then are all DLWP models' degradation because of autoregressive fine-tuning? This seems like a crucial problem to just hypothesize the cause. I think it would be beneficial for readers to pinpoint the cause.

Some papers have suggested that models with multistep fine-tuning have blurry results [Lang et al., 2024, Brenowitz et al., 2024], in part because they try to predict small-scale features beyond the predictability horizon [Kochkov et al., 2024] In our paper, we perform the converse experiment: SFNO trained without multistep fine-tuning largely does not get increasingly blurry during the rollout. This was an important factor in our choice not to multistep fine-tune: many other data-driven weather prediction models do choose to perform multistep fine-tuning for improved benchmark scores on overall diagnostics. However, because many DDWPs have very different architectures, time integrators, weighted loss functions, input variable sequences (such as including the most recent timestep or the most recent two timesteps), and targets (such as predicting the full atmospheric state or the residual between the input atmospheric state and the next step) we cannot exactly pinpoint the cause for all architectures in this work.

7. (p.19) "This is necessary but as yet insufficient validation for our main scientific interest in LLHIs.": I expect more analysis of LLHIs such as case studies that occurred during recent years, even though the authors agreed with the lack of validation. It would provide a more robust evaluation and help illustrate the model's practical value.

Thank you for these suggestions on improving the rigor of our manuscript. We will add these case studies. The context of this sentence referred to the ability of overall diagnostics (CRPS, ensemble mean RMSE) to assess performance on extreme events specifically. We introduced extreme diagnostics in Section 3.3 as our primary means of validation on extremes. These diagnostics assess statistical performance and reduce reliance on anecdotal evidence. However, we agree that case studies and examples are valuable ways to illustrate the model's practical value, and we will build upon the case studies in Part II and add an example for wind events. Thank you for your suggestion.

Minor Comments

1. (p.12) "First, they contain a land-sea contrast for surface fields such as 10m wind speed and 2m temperature. For these surface fields, perturbations have distinct amplitudes and spatial scales over the land and ocean.": It's a bit difficult for me to discriminate the difference. Could you show the amplitude in

another way?

Thank you for pointing out this issue to us. We appreciate your feedback. We will modify the figure and its caption in our revised manuscript to help clarify this.

2. (p.14) "On 850 hPa temperature, 2m temperature, 850 hPa specific humidity, and 500 hPa geopotential, SFNO-BVMC lags approximately 18 hours behind IFS ENS, though their performance is comparable.": CRPS score with all pressure levels would be useful for readers e.g. GenCast or GraphCast.

We appreciate this suggestion. Even with a 58-member ensemble (not the huge ensemble), validating on all pressure levels used in the model would require a significant expense. This is a calculation on approximately 685 terabytes of data: (721: lat x 1440: lon x 60: lead time x 365: initial days x 58: ensemble members x 13: pressure levels x 5: prognostic variables x 32 bits). We could reduce the number of initial days on which we validate, but this would still be a sizable task. We agree that validating on all pressure levels would provide interesting information. However, such validation is not central to the scientific core of our two papers. For the sake of brevity of our already two-part manuscript, and to be judicious with our resources, we suggest saving this validation for future work. We note that the model weights are already made available under the very open CC0 license. If an interested reader seeks to use our ensemble for a different purpose than considered here (for example, forecasting 50 hPa variables), then they can run and validate the model openly for this task.

**References**

M. Alexe, S. Lang, M. Clare, M. Leutbecher, C. Roberts, L. Magnusson, M. Chantry, R. Adeowyin, A. Prieto-Nemesio, J. Dramsch, F. Pinault, and B. Raoult. Data-driven ensemble forecasting with the AIFS . https://www.ecmwf.int/en/newsletter/181/earth-system-science/data-driven-ensemble-forecasting-aifs, 2024. [Accessed 17-02-2025]. 2

K. Bi, L. Xie, H. Zhang, X. Chen, X. Gu, and Q. Tian. Accurate medium-range global weather forecasting with 3D neural networks. *Nature*, 619(7970):533–538, July 2023. ISSN 1476-4687. doi:10.1038/s41586-023-06185-3. URL http://dx.doi.org/10.1038/s41586-023-06185-3. 6

C. Bodnar, W. P. Bruinsma, A. Lucic, M. Stanley, A. Vaughan, J. Brandstetter, P. Garvan, M. Riechert, J. A. Weyn, H. Dong, J. K. Gupta, K. Thambiratnam, A. T. Archibald, C.-C. Wu, E. Heider, M. Welling, R. E. Turner, and P. Perdikaris. A foundation model for the earth system, 2024. URL https://arxiv.org/abs/2405.13063. 6

M. Bonavita. On Some Limitations of Current Machine Learning Weather Prediction Models. *Geophysical Research Letters*, 51(12), June 2024. ISSN 1944-8007. doi:10.1029/2023gl107377. URL http://dx.doi.org/10.1029/2023GL107377. 8

B. Bonev, T. Kurth, C. Hundt, J. Pathak, M. Baust, K. Kashinath, and A. Anandkumar. Spherical Fourier Neural Operators: Learning stable dynamics on the sphere. In *International conference on machine learning*, pages 2806–2823. PMLR, 2023. 6

N. D. Brenowitz, Y. Cohen, J. Pathak, A. Mahesh, B. Bonev, T. Kurth, D. R. Durran, P. Harrington, and M. S. Pritchard. A practical probabilistic benchmark for AI weather models. *arXiv preprint arXiv:2401.15305*, 2024. 2, 8

S. R. Cachay, B. Henn, O. Watt-Meyer, C. S. Bretherton, and R. Yu. Probabilistic Emulation of a Global Climate Model with Spherical DYffusion. *arXiv preprint arXiv:2406.14798*, 2024. 6

K. Chen, T. Han, J. Gong, L. Bai, F. Ling, J.-J. Luo, X. Chen, L. Ma, T. Zhang, R. Su, et al. FengWu: Pushing the Skillful Global Medium-range Weather Forecast beyond 10 Days Lead. *arXiv preprint arXiv:2304.02948*, 2023a. 6

L. Chen, X. Zhong, F. Zhang, Y. Cheng, Y. Xu, Y. Qi, and H. Li. FuXi: A cascade machine learning forecasting system for 15-day global weather forecast. *npj Climate and Atmospheric Science*, 6(1):190, 2023b. doi:10.1038/s41612-023-00512-1. URL https://doi.org/10.1038/s41612-023-00512-1. 6

C. Johnson and N. Bowler. On the Reliability and Calibration of Ensemble Forecasts. *Monthly Weather Review*, 137(5):1717–1720, May 2009. ISSN 0027-0644. doi:10.1175/2009mwr2715.1. URL http://dx.doi.org/10.1175/2009MWR2715.1. 7

R. Keisler. Forecasting global weather with graph neural networks. *arXiv preprint arXiv:2202.07575*, 2022. 6

D. Kochkov, J. Yuval, I. Langmore, P. Norgaard, J. Smith, G. Mooers, M. Klöwer, J. Lottes, S. Rasp, P. Düben, S. Hatfield, P. Battaglia, A. Sanchez-Gonzalez, M. Willson, M. P. Brenner, and S. Hoyer. Neural general circulation models for weather and climate. *Nature*, 632(8027):1060–1066, July 2024. ISSN 1476-4687. doi:10.1038/s41586-024-07744-y. URL http://dx.doi.org/10.1038/s41586-024-07744-y. 8

R. Lam, A. Sanchez-Gonzalez, M. Willson, P. Wirnsberger, M. Fortunato, F. Alet, S. Ravuri, T. Ewalds, Z. Eaton-Rosen, W. Hu, et al. Learning skillful medium-range global weather forecasting. *Science*, 382 (6677):1416–1421, 2023. doi:10.1126/science.adi2336. URL https://www.science.org/doi/abs/10.1126/science.adi2336. 6

S. Lang, M. Alexe, M. Chantry, J. Dramsch, F. Pinault, B. Raoult, M. C. A. Clare, C. Lessig, M. Maier-Gerber, L. Magnusson, Z. B. Bouallègue, A. P. Nemesio, P. D. Dueben, A. Brown, F. Pappenberger, and F. Rabier. AIFS – ECMWF's data-driven forecasting system, 2024. URL https://arxiv.org/abs/2406.01465. 8

D. A. Lavers, F. Pappenberger, D. S. Richardson, and E. Zsoter. ECMWF Extreme Forecast Index for water vapor transport: A forecast tool for atmospheric rivers and extreme precipitation. *Geophysical Research Letters*, 43(22), Nov. 2016. ISSN 1944-8007. doi:10.1002/2016gl071320. URL http://dx.doi.org/10.1002/2016GL071320. 4

A. Mahesh, W. Collins, B. Bonev, N. Brenowitz, Y. Cohen, J. Elms, P. Harrington, K. Kashinath, T. Kurth, J. North, et al. Huge Ensembles Part I: Design of Ensemble Weather Forecasts using Spherical Fourier Neural Operators. *arXiv preprint arXiv:2408.03100*, 2024a. 1

A. Mahesh, W. Collins, B. Bonev, N. Brenowitz, Y. Cohen, P. Harrington, K. Kashinath, T. Kurth, J. North, T. OBrien, et al. Huge Ensembles Part II: Properties of a Huge Ensemble of Hindcasts Generated with Spherical Fourier Neural Operators. *arXiv preprint arXiv:2408.01581*, 2024b. 1

J. Pathak, S. Subramanian, P. Harrington, S. Raja, A. Chattopadhyay, M. Mardani, T. Kurth, D. Hall, Z. Li, K. Azizzadenesheli, et al. Fourcastnet: A global data-driven high-resolution weather model using adaptive fourier neural operators. *arXiv preprint arXiv:2202.11214*, 2022. 6

343 I. Price, A. Sanchez-Gonzalez, F. Alet, T. R. Andersson, A. El-Kadi, D. Masters, T. Ewalds, J. Stott, S. Mo-
344 hamed, P. Battaglia, R. Lam, and M. Willson. Probabilistic weather forecasting with machine learn-
345 ing. *Nature*, 637(8044):84–90, Dec. 2024. ISSN 1476-4687. doi:10.1038/s41586-024-08252-9. URL
346 http://dx.doi.org/10.1038/s41586-024-08252-9. 6

347 V. Ramavajjala. HEAL-ViT: Vision Transformers on a spherical mesh for medium-range weather forecast-
348 ing. *arXiv preprint arXiv:2403.17016*, 2024. 6

**Second Response to Reviewers: "Huge Ensembles Part I: Design of Ensemble Weather Forecasts using Spherical Fourier Neural Operators"**

April 3, 2025

**Overview**

We sincerely appreciate the reviewers' thoughtful feedback and thorough evaluation of our paper. In our prior response (https://egusphere.copernicus.org/preprints/2024/egusphere-2024-2420/egusphere-2024-2420-AC2-supplement.pdf), we included our outlined plan for changing the manuscript (e.g. changing text, adding new figures, updating existing figures). We have now submitted our revised manuscript. In this document, we include pointers to the updated manuscripts that resolve the reviewers' major comments: we do not include all the reviewers' comments here. For a complete line-by-line discussion of all the reviewers' comments, please see our prior response at the link above. Our manuscript changes are in green, and the reviewer comments are in black.

**Changes to the Manuscript In Response to Reviewer #1**

Page 2: The ML model has "orders-of-magnitudes" lower computational cost. Is this really true? More than a factor of 10? This could only be possible if the IO cost (that will stay the same) is considered to be of less than 10% of the overall cost (also see comment for Part 2). And what is the "cost"? Time, energy, or hardware purchase?

See the Part II changes, where we add a sentence on the computational efficiency of ML that makes it feasible to generate 256 members in one minute in parallel on 256 GPUs.

P6, paragraph starting with "We choose SFNO...": I find this part difficult to follow. It would be good to remind the reader about the architecture of the SFNO.

We have now included a description of the SFNO architecture in section 2.1

What exactly do you mean by downscaling and scale factor here (I think I know, but only since I know the previous papers)? I do not understand why a lower scale factor would lead to a larger ensemble spread.

We added this to the architecture description in Section 2.1. Also, on page 7, we have included the following text:

*The scale factor controls the level of spectral downsampling of the input field. With more aggressive downsampling, SFNO internally represents the input atmospheric state with reduced resolution. We speculate*

*that this may reduce the effective resolution of the predictions. With a reduced effective resolution, small-scale perturbations would not grow and propagate upscale. Instead, they would be blurred out, and they would not result in increased spread among ensemble members.*

Figure 10: Is the control member equivalent to a normal ensemble member, or are there small differences (as in IFS)? Can you also plot 0h?

See Figure D5. Also, see associated text:

*At a lead time of 360 hours, the perturbed members maintain similar spectra as the control member (Figure 10), and at the initial time, they have similar spectral characteristics as the unperturbed ERA5 initial condition (Figure D5).*

Figure 2b: For what timestep are the spectra calculated?
They are calculated at 360 hours. We have added this information to the figure caption for Figure 2b.

**Changes to the Manuscript In Response to Reviewer #2**

Despite these promising results, several aspects warrant further research. The authors mainly focused on 2m temperature, especially heat extreme from model configuration to diagnostics, and given that the authors deliberately included 2m dewpoint temperature as a model input variable, incorporating predictions of derived heat extreme indices would provide valuable insights into the model's capabilities.

We have included diagnostics on the heat index in Appendix D, Figure D2

Furthermore, I recommend evaluating a broader range of LLHIs to strengthen the reliability of the approach. I think the authors can incorporate cold extremes along with heat extremes. What about wind extremes, which are in prediction variables?

We have included diagnostics on the 10m wind speed and cold extremes in Appendix D, Figure D3 and Figure D4. In the appendix and in the main text, we state that the model performs well on these other variables at 48 and 96 hours. At 240 hours, the model's reliability degrades for probabilities greater than approximately 50%. This is the subject for further research. In addition to appendix D, we highlight this in the main text also on page 23:

*We visualize the reliability diagrams for other lead times (Supplemental Figure D1) and variables. We show that SFNO-BVMC also performs reliably when forecasting the heat index at lead times of 48, 96, 120, and 240 hours. For 10m wind speed and cold extremes, SFNO-BVMC matches the performance of the IFS ensemble (Figure D2 and Figure D3). However, we also show that at 240 hour lead times, the model is not reliable when it confidently (greater than 50% chance) forecasts wind extremes or cold temperature extremes (see Appendix D and Figure D4 for more discussion). This is an area for future model development.*

As well as various extreme events, actual forecasts would be helpful to recognize the usefulness of the model. Diagnostics with real-event prediction would be more persuasive. For example, t2m ensemble time series at a certain grid point, trajectories of each ensemble for each variable, and the difference between IFS could strengthen the model's credibility.

We have included a real-event prediction demo in Appendix A.

Major Comments 1. (p.4) "Existing work has shown that simple Gaussian perturbations do not yield a sufficiently dispersive ensemble. (Scher and Messori, 2021; Bülte et al., 2024): the ensemble spread from these perturbations is too small." If so, you can still adopt singular vectors or other methods to reflect initial condition uncertainty. Are bred vectors superior to other approaches? Are they the cheapest way other than simple Gaussian perturbations?

Thank you very much for raising this helpful suggestion. We have added this to the discussion section, page 26:

*Understanding how ML models respond to perturbations is an important research frontier. In particular, future work is necessary to compare the computational cost and skill of different initial condition perturbation methods, in tandem with model perturbations. We find that bred vectors are a computationally inexpensive way to achieve reasonable spread-error ratios and to generate an arbitrarily large ensemble. Further refinement of initial condition perturbation techniques is needed to improve forecast performance.*

We continue the discussion by comparing bred vectors to other perturbation methods.

4. (p.10) Figure 3. The ensemble spread from different numbers of checkpoints. : Model configuration also focused on 2m temperature. Do we need to change the number of checkpoints if we want to forecast wind extremes? Do we need to change it every time for different variables? Selecting the number of checkpoints based on the comparison among multiple variables would be a more optimal choice.

We have added 2 additional variables to our analysis to Figure 3.

5. (p.17) "On the second criterion, crucially, their spectra remain constant through the 360-hour rollout (Figure 10 and Figure 11)."
: Degradation of power in short wavelengths occurs in a lot of DLWPs. Then are all DLWP models' degradation because of autoregressive fine-tuning? This seems like a crucial problem to just hypothesize the cause. I think it would be beneficial for readers to pinpoint the cause.
6. (p.17) "While the control and perturbed spectra remain constant through the rollout, the SFNO-BVMC ensemble mean does increasingly blur with lead time. Figure 12 shows that the ensemble means of SFNO-BVMC and IFS ENS similarly degrade in power after 24 hours, 120 hours, and 240 hours."
: In the first paragraph of section 3.2 Spectral Diagnostics, the authors elaborate that power decay is one of the symptoms of blurriness, but this sentence seems like presuming those two are equivalent. section 3.2 needs to be more clear. What is the relationship between spectra and blurriness in general and what did SFNO find? Why is SFNO-BVMC different from other DLWPs with respect to the power spectrum?

We have completely rewritten "Section 3.2 Spectral Diagnostics" on page 17 to address these questions.

7. (p.19) "This is necessary but as yet insufficient validation for our main scientific interest in LLHIs."
: I expect more analysis of LLHIs such as case studies that occurred during recent years, even though the authors agreed with the lack of validation. It would provide a more robust evaluation and help illustrate the model's practical value.

We have included a case study in appendix A.

Minor Comments
1. (p.12) "First, they contain a land-sea contrast for surface fields such as 10m wind speed and 2m

temperature. For these surface fields, perturbations have distinct amplitudes and spatial scales over the land and ocean.": It's a bit difficult for me to discriminate the difference. Could you show the amplitude in another way?

We have made the following change:

*For these surface fields, the perturbations have distinct amplitudes over the land and ocean. In this example, the 2m temperature perturbation has an amplitude of 0.56 K over land and 0.27 K over the ocean, and the 10m wind speed perturbation has an amplitude of 0.45 m/s over land and 0.66 m/s over the ocean.*

**References**